# Chromosome mis-segregation and cytokinesis failure in trisomic human cells

Joshua M Nicholson[1,2], Joana C Macedo[3,4,5], Aaron J Mattingly[1†], Darawalee Wangsa[6], Jordi Camps[6‡], Vera Lima[7], Ana M Gomes[3], Sofia Dória[7], Thomas Ried[6], Elsa Logarinho[3,4,5]*, Daniela Cimini[1,2]*

[1]Department of Biological Sciences, Virginia Tech, Blacksburg, United States; [2]Virginia Bioinformatics Institute, Virginia Tech, Blacksburg, United States; [3]Aging and Aneuploidy Laboratory, Instituto de Biologia Molecular e Celular, Universidade do Porto, Porto, Portugal; [4]Instituto de Investigação e Inovação em Saúde-i3S, Universidade do Porto, Porto, Portugal; [5]Cell Division Unit, Department of Experimental Biology, Faculdade de Medicina, Universidade do Porto, Porto, Portugal; [6]Genetics Branch, National Cancer Institute, National Institutes of Health, Bethesda, United States; [7]Department of Genetics, Faculdade de Medicina, Universidade do Porto, Porto, Portugal

*For correspondence: elsa.
logarinho@ibmc.up.pt (EL);
cimini@vt.edu (DC)

Present address: †Cell and Tissue
Biology, School of Dentistry,
University of California, San
Francisco, San Francisco, United
States; ‡Gastrointestinal and
Pancreatic Oncology Group,
Hospital Clínic, Centro de
Investigación Biomédica en Red
de Enfermedades Hepáticas y
Digestivas, Institut
D'Investigacions Biomèdiques
August Pi i Sunyer, Barcelona,
Spain

Competing interests: The
authors declare that no
competing interests exist.

Gurdon Institute, United
Kingdom

**Abstract** Cancer cells display aneuploid karyotypes and typically mis-segregate chromosomes at high rates, a phenotype referred to as *chromosomal instability* (CIN). To test the effects of aneuploidy on chromosome segregation and other mitotic phenotypes we used the colorectal cancer cell line DLD1 (2n = 46) and two variants with trisomy 7 or 13 (DLD1+7 and DLD1+13), as well as euploid and trisomy 13 amniocytes (AF and AF+13). We found that trisomic cells displayed higher rates of chromosome mis-segregation compared to their euploid counterparts. Furthermore, cells with trisomy 13 displayed a distinctive cytokinesis failure phenotype. We showed that up-regulation of SPG20 expression, brought about by trisomy 13 in DLD1+13 and AF+13 cells, is sufficient for the cytokinesis failure phenotype. Overall, our study shows that aneuploidy can induce chromosome mis-segregation. Moreover, we identified a trisomy 13-specific mitotic phenotype that is driven by up-regulation of a gene encoded on the aneuploid chromosome.

## Introduction

Aneuploidy, an abnormal number of chromosomes, is a leading cause of mis-carriage and birth defects in humans (*Nagaoka et al., 2012*). In the vast majority of cases, this is due to errors occurring in the oocyte (*Nagaoka et al., 2012*). However, aneuploidy can also arise in somatic cells, and a number of studies have reported age-dependent increases in aneuploidy in human peripheral blood lymphocytes (*Nowinski et al., 1990*; *Carere et al., 1999*; *Leopardi et al., 2002*). Moreover, aneuploidy was recognized as a common feature of cancer cells already a century ago (*Boveri, 1914*, *2008*), and a causal role of aneuploidy in carcinogenesis is currently largely acknowledged (reviewed in [*Pavelka et al., 2010a*; *Nicholson and Cimini, 2011*]). In addition to being aneuploid, cancer cells typically display high rates of chromosome mis-segregation, a phenomenon termed *chromosomal instability* (CIN) (*Lengauer et al., 1997*; *Bakhoum et al., 2014*). The observation that even mosaic aneuploidy can cause severe physical and cognitive developmental defects (*Biesecker and Spinner, 2013*) indicates that aneuploidy has pleiotropic deleterious effects. This idea is further supported by a number of experimental observations: first, knocking down spindle assembly checkpoint genes, which results in high rates chromosome mis-

**eLife digest** The DNA in a human cell is divided between forty-six structures called chromosomes. Before a cell divides, it copies every chromosome so that each daughter cell will have the same DNA as the parent cell. These chromosomes align in the center of the cell and then the matching chromosomes are separated and pulled to opposite ends.

However, in some cases the separation process does not work properly, which can produce cells that either have too many, or too few, chromosomes. Abnormal numbers of chromosomes within cells—called aneuploidy—is a leading cause of miscarriage and birth defects in humans. Aneuploidy is also a common feature of cancer cells.

It is common for the chromosomes in cancer cells to be distributed unequally when the cell divides. This phenomenon is known as chromosomal instability, but the link between aneuploidy and chromosomal instability in cancer cells is not fully understood.

Here, Nicholson et al. used live-cell imaging techniques to analyze healthy human cells and cancer cells that had either the normal forty-six chromosomes, or a defined extra chromosome. Nicholson et al. found that when the cells divided, the chromosomes in the cells that had an extra copy of chromosome 7 or 13 were more prone to distributing chromosomes unequally, compared to cells with a normal number of chromosomes.

Nicholson et al. also observed that the cells with an extra chromosome 13 were unable to properly divide into two. These cells had increased levels of a protein called Spartin—which is important for the last stage in cell division—and this was responsible for the failure to produce two daughter cells.

These findings show that aneuploidy can cause chromosomal instability in human cells. Furthermore, Nicholson et al. have identified a defect in cell division that is specifically caused by the presence of an extra chromosome 13 in human cells. A future challenge will be to determine how, and to what extent, different chromosomes can affect chromosome stability. This could be useful in the development of therapies against cancer cells with aneuploidy.

segregation and high levels of aneuploidy, invariably causes embryonic lethality in mouse models (*Foijer et al., 2008*). Second, aneuploid yeast strains were shown to exhibit defects in cell cycle progression and metabolism (*Torres et al., 2007*). Third, MEFs derived from mice carrying specific trisomies were found to display cell proliferation defects and metabolic alterations (*Williams et al., 2008*). Finally, genes involved in stress response were shown to be upregulated in aneuploid yeast and human cells (*Sheltzer et al., 2012*; *Stingele et al., 2012*). But in the context of cancer, aneuploidy and CIN strongly correlate with drug resistance (*Lee et al., 2011*) and poor patient prognosis (*Bardi et al., 2004*; *Carter et al., 2006*; *Walther et al., 2008*; *Sheffer et al., 2009*), indicating that aneuploidy and CIN may confer a proliferative advantage to cancer cells. In support of this idea, certain aneuploidies were found to confer drug resistance in aneuploid *Saccharomyces cerevisiae* (*Pavelka et al., 2010b*) and *Candida albicans* (*Selmecki et al., 2006*, *2009*). These studies suggest that aneuploidy may confer adaptability by inducing chromosome-specific phenotypic changes, despite general negative effects on cell physiology. However, this problem remains to be investigated in human cells.

Recent work in aneuploid budding yeast also showed that aneuploidy is sufficient to cause CIN (*Sheltzer et al., 2011*; *Zhu et al., 2012*), but whether this is true in human cells is still a matter of debate (*Duesberg, 2014*; *Heng, 2014*; *Valind and Gisselsson, 2014a*, *2014b*). In fact, this question has been difficult to address in cancer cells due to the complexity of cancer karyotypes (*Gisselsson, 2011*; *Mitelman et al., 2014*), and previous studies in human cancer and non-cancer cells have reached discrepant conclusions (*Lengauer et al., 1997*; *Duesberg et al., 1998*; *Miyazaki et al., 1999*; *Valind et al., 2013*). To determine the effect of aneuploidy on chromosome segregation and cell division in human cells, we utilized a number of diploid human cell types and trisomic counterparts, including: colorectal cancer cell line DLD1 (2n = 46) and trisomic counterparts carrying extra copies of chromosomes 7 or 13 (DLD1+7 and DLD1+13, respectively); diploid amniotic fibroblasts (AF) and amniotic fibroblasts with trisomy 13 (AF+13). These different cell types constitute a good model for our study for two main reasons: first, their karyotypes are aneuploid, but not as complex as typically found in tumors and cancer cell lines; second, they represent different cellular models (transformed and untransformed) of aneuploidy.

## Results

DLD1+7 and DLD1+13 cell lines were previously generated by micro-cell mediated chromosome transfer (*Upender et al., 2004*), whereas AF and AF+13 cells (three cases each; see *Table 1*) were collected upon amniocentesis. The presence of the additional chromosome was confirmed by fluorescence in situ hybridization (FISH) with locus-specific probes (*Figure 1A–B*). Analysis of DLD1+7 cells previously showed that a large fraction (87%) of the population was trisomic (*Upender et al., 2004*). However, the DLD1+13 cell population was shown to rapidly accumulate disomic (by loss of one copy of chromosome 13) and tetraploid cell populations (*Upender et al., 2004*). Thus, for this study we sub-cloned DLD1+13 cells in order to select a more homogenous cell population. When we analyzed the clone selected for this study at early passages (P. 3–4) by chromosome 13 painting, we found that 83.5% of the cells in the population carried the trisomy 13 (*Figure 1C*). Similarly, analysis of AF+13 interphase nuclei (passage 1–2) FISH-stained with probes specific for chromosomes 13 and 21 showed that the cell populations used in this study were highly homogenous (88.1 ± 6.5%) for the trisomic karyotype (*Figure 1C*). Furthermore, we performed array comparative genomic hybridization (aCGH) of all three DLD1 cell lines (*Figure 1—figure supplement 1A,B,E*). In all DLD1 cell lines, we found amplification of regions on the p arm of chromosomes 2 and 11 and a deletion of a region on the p arm of chromosome 6, which are known to be recurrently found in DLD1 cells. In addition to these common copy number variations (CNVs), the DLD1+7 cell line (analyzed at passage 4) carried a partial trisomy 7 including most of the q arm (*Figure 1—figure supplement 1B–C*). FISH staining with a probe specific to the centromere of chromosome 7 confirmed that the extra chromosome included a centromere (*Figure 1—figure supplement 1D*). aCGH of DLD1+13 cells (at passage 11) showed that in addition to the CNVs identified in all three DLD1 cell lines, there was an extra copy of the entire chromosome 13 (*Figure 1—figure supplement 1E–F*). The experiments described hereafter were performed at passage number 7–25 for DLD1+7 cells and 13–25 for DLD1+13 cells to limit evolution of the karyotypes and passage number 1–3 for amniocytes, whose proliferation was limited to few passages.

### Increased chromosome mis-segregation in cells with trisomy 7 or 13

To investigate the effect of aneuploidy on chromosome segregation, we analyzed anaphase lagging chromosomes, a common cause of aneuploidy in normal and cancer cells (*Cimini et al., 2001*; *Thompson and Compton, 2008*). By analyzing fixed cells with immunostained kinetochores and microtubules, we found that DLD1+7 and DLD1+13 cells displayed significantly higher frequencies of anaphase lagging chromosomes compared to the parental DLD1 cell line (*Figure 2A–B*). We found no evidence of aneuploidy-dependent increases in other mitotic defects, such as multipolar mitoses and anaphase chromosome bridges (*Figure 2—figure supplement 1*). Frequencies of anaphase lagging chromosomes in AF and AF+13 cells could not be analyzed in fixed samples due to low mitotic indices. However, we optimized live-cell imaging of AF and AF+13 cells expressing H2B-GFP/RFP-tubulin (*Figure 2C*; *Videos 1–2*) and found higher frequencies of anaphase lagging chromosomes in AF+13 compared to AF cells (*Figure 2D*). Anaphase chromosome bridges or multipolarity were never observed in AF or AF+13 cells.

As an additional method to measure chromosome mis-segregation and to account for events in which two sister chromatids co-segregate to the same spindle pole/daughter cell, we combined the cytokinesis-block assay (*Fenech, 1993*) with FISH staining using locus-specific probes for chromosomes 3, 7, 11, and 13 in DLD1 cell lines, and probes specific for chromosomes 7, 11, 12, 13, 18, and 19 in amniocytes. Using this approach, which allows analysis of the reciprocal distribution of chromosomes between the daughter nuclei of a single mitotic division (*Figure 3A*, *Figure 3—figure supplement 1*), we found a significant increase in chromosome mis-segregation in DLD1+7, DLD1+13, and AF+13 compared to the corresponding diploid cell

**Table 1**. Euploid and trisomic amniocytes used in this study

| Name | Karyotype | Gestational age (week) |
|---|---|---|
| AF 1 | 46,XX | 17 |
| AF 2 | 46,XY | 20 |
| AF 3 | 46,XX | 21 |
| AF+13 1 | 47,XX,+13 | 16 |
| AF+13 2 | 47,XX,+13 | 21 |
| AF+13 3 | 47,XX,+13 | 21 |
| AF+18 | 47,XX,+18 | 17 |
| AF+21 | 47,XX,+21 | 17 |

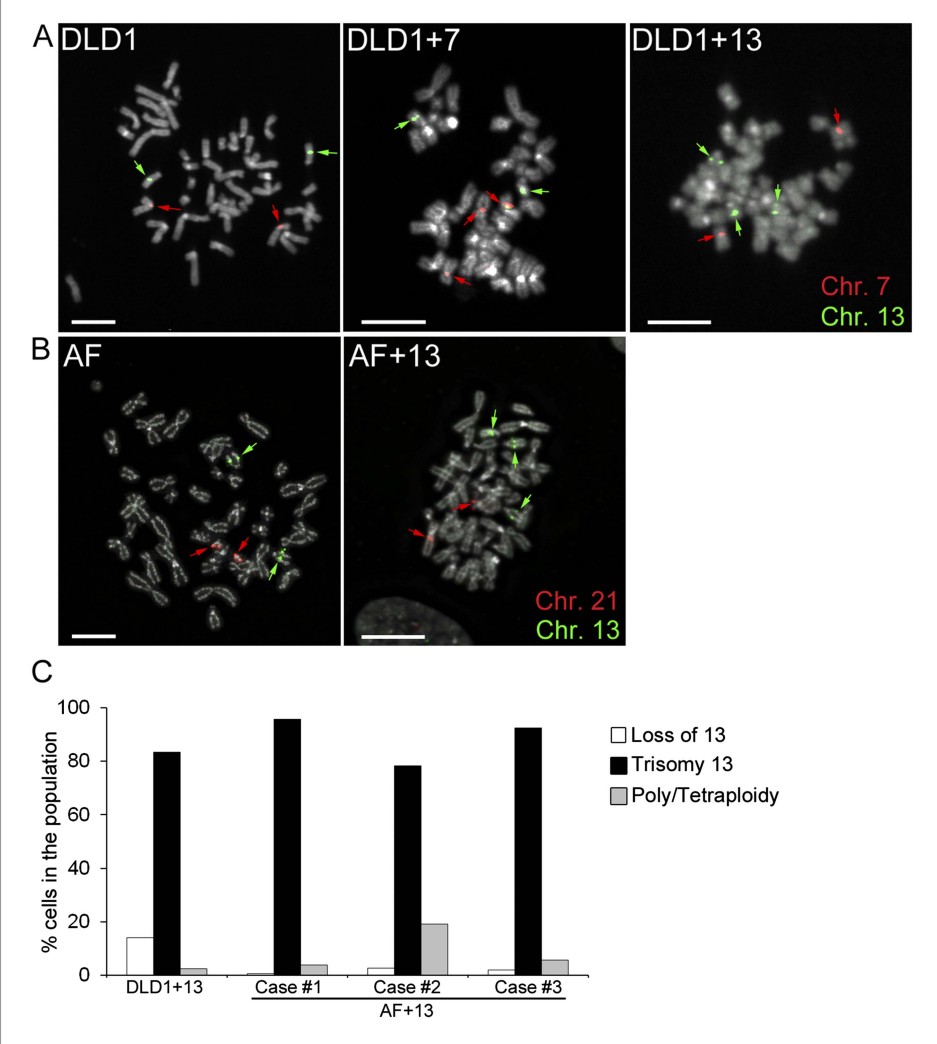

**Figure 1**. Trisomy 7 and 13 in DLD1 and AF cells. (**A–B**) Chromosome-specific FISH staining for chromosomes 7 (red),13 (green), and 21 (red) in metaphase spreads confirms trisomy 7 in DLD1+7 and trisomy 13 in DLD1+13 and AF+13 cells. DNA is shown in grey. Arrows point to FISH signals. Scale bars, 5 µm. (**C**) Characterization of cells with trisomy 13 used in this study. The DLD1+13 cells were subcloned and the clone used for this study was analyzed by chromosome painting of metaphase spreads with chromosome 13-specific probes. Cells were classified as having lost chromosome 13, trisomic for chromosome 13, or poly/tetraploid when they carried six copies of chromosome 13 and ~88 other chromosomes. The AF+13 cells were analyzed by interphase FISH at passage 1–2 with probes specific to chromosomes 13 and 21. Cells were classified has having lost/gained a copy of chromosome 13 (two or four nuclear signals for chr. 13, two signals for chr. 21), displaying trisomy 13 (three signals for chr. 13 and 2 signals for chr. 21), or displaying poly/tetraploidy (six signals for chr. 13 and four signals for chr. 21). A small fraction (1.9%) of the AF+13 Case #3 displayed loss/gain of chr. 21.

The following figure supplement is available for figure 1:

**Figure supplement 1**. aCGH in DLD1, DLD1+7, and DLD1+13 cells.

cultures (*Figure 3B–C*). However, the higher mis-segregation rates were specific to certain chromosomes. Namely, mis-segregation appeared to be increased for chromosome 7 in the DLD1+7 cell line, chromosomes 7 and 13 in DLD1+13 cells, and chromosome 13 in AF+13 cells as compared to their diploid counterparts. We further investigated whether the observed increases in chromosome mis-segregation rates impacted chromosome number variability (or karyotypic heterogeneity) in the trisomic cells compared to the diploid counterparts. To this end, we performed chromosome counts in

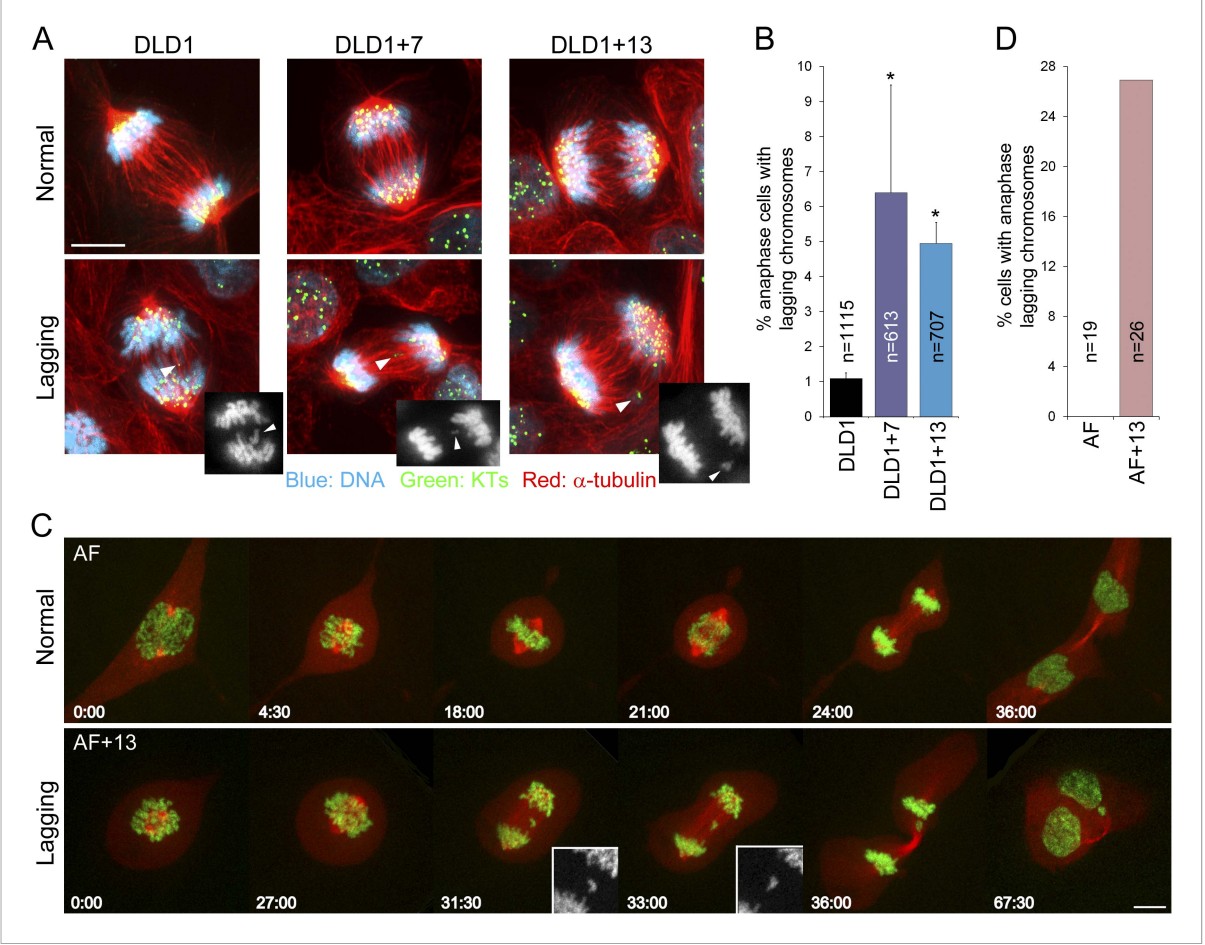

Blue: DNA   Green: KTs   Red: α-tubulin

**Figure 2**. Increased rates of anaphase lagging chromosomes in cells with trisomy 7 or 13. (**A**) Examples of normal anaphases (top row) and anaphase cells with lagging chromosomes (bottom row). Cells were immunostained for microtubules (red) and kinetochores (green). DNA is shown in blue. Images represent maximum intensity projections of Z-stacks. Arrowheads point at anaphase lagging chromosomes. Grey scale images at the bottom right corners of the images in the bottom row are single focal planes of DAPI-stained chromosomes shown for easier visualization of the lagging chromosomes. (**B**) Frequencies of anaphase lagging chromosomes were significantly higher (*$\chi^2$ test, p < 0.0001) in both DLD1+7 and DLD1+13 compared to DLD1 cells. Data are reported as mean ± S.E.M and represent the average of three independent experiments in which a total of 613–1115 anaphases were analyzed. (**C**) Time-lapse microscopy of AF and AF+13 cells undergoing mitosis. An example of AF undergoing normal mitosis is shown in the top row and an example of AF+13 displaying an anaphase lagging chromosome is shown in the bottom row. DNA is shown in green (H2B-GFP) and microtubules in red (RFP-tubulin). Images are maximum intensity projections of Z-stacks. Insets in the bottom row display enlarged views of the DNA alone (in grey scale) in the region around the lagging chromosome. (**D**) None of the 19 AF cells (from cases #1 and #2) imaged displayed anaphase lagging chromosomes, whereas 7 out of 26 AF+13 cells (5 out of 18 in case #1 and 3 out of 8 in case #2) displayed anaphase lagging chromosomes. Time stamps indicate elapsed time in min:sec. Scale bars, 5 μm.

The following figure supplement is available for figure 2:

**Figure supplement 1**. Similar frequencies of multipolar mitoses (**A**) and anaphase chromosome bridges (**B**) in diploid vs aneuploid DLD1 cells.

metaphase spreads and found increased karyotypic heterogeneity in both DLD1+7 and DLD1+13 compared to DLD1 cells. We also found higher karyotypic heterogeneity in AF+13 vs AF cells around the modal chromosome number of 47 (*Figure 3D*). Finally, our chromosome counts revealed the presence of a tetraploid/near-tetraploid sub-population in DLD1+13 cells (*Figure 3D*), confirming previous findings (*Upender et al., 2004*). Because chromosome counts did not reveal a significant difference in tetraploid cells between AF and AF+13 cell populations (*Figure 3D*), we decided to further characterize chromosome number variability in these cells by performing FISH analysis with locus-specific probes for chromosomes 7, 12, and 18 on interphase nuclei (*Figure 3E*), which allowed for larger numbers of cells to be examined. This analysis confirmed higher degrees of aneuploidy in

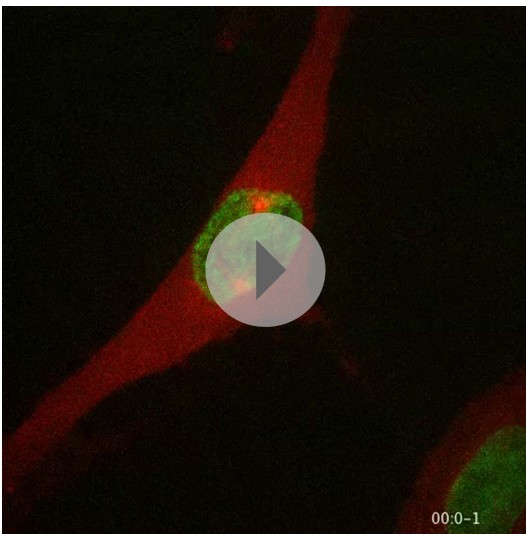

**Video 1.** Representative video showing normal mitosis in an AF cell. Images were acquired by spinning-disk confocal microscopy at 1 min intervals and they are played back at 5 frames per second. DNA is shown in green (H2B-GFP) and microtubules in red (RFP-tubulin).

AF+13 compared to AF cells (*Figure 3F*). Furthermore, it revealed the presence of a tetraploid sub-population in AF+13 cells (*Figure 3F*). The difference between numbers of AF+13 tetraploid interphase cells and tetraploid chromosome spreads (compare *Figure 3D* and *Figure 3F*) may be due to the inability of tetraploid AF+13 cells to re-enter mitosis (see also 'Discussion' section). Taken together, these experiments (*Figures 2–3*) show that trisomies 7 and 13 cause chromosome mis-segregation, that mis-segregation affects certain chromosomes more than others, and that such increases in mis-segregation rates are associated with karyotypic heterogeneity within the cell population. However, because only trisomies 7 and 13 were examined, and a limited number of chromosomes analyzed in our FISH experiments, it remains elusive whether chromosome mis-segregation is a karyotype-specific or a general effect of aneuploidy in human cells.

## Trisomy 13 promotes cytokinesis failure due to the overexpression of SPG20

Our chromosome counts (*Figure 3D*) showed that DLD1+13 cells displayed a near-tetraploid sub-population, which was also evident in our FISH experiments in which nuclei with four or more signals per chromosome were more frequent in DLD1+13 compared to DLD1 cells (24.5% vs 16.6%, $\chi^2$, p < 0.0001). Similarly, a tetraploid sub-population was evident in AF+13 cells (4.5% vs 0.3% in AF, $\chi^2$, p < 0.0001) analyzed by interphase FISH, which

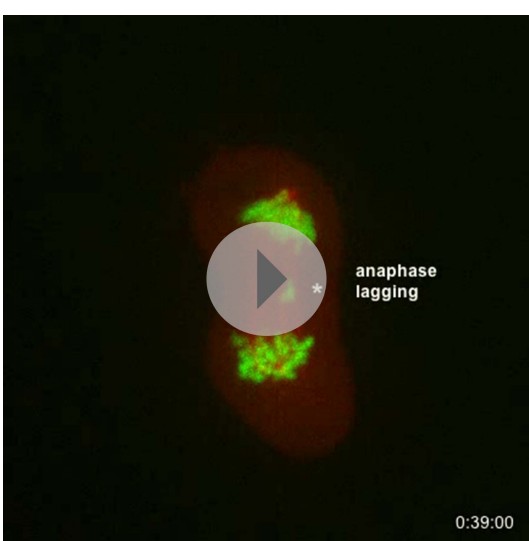

**Video 2.** Representative video showing anaphase lagging chromosome during mitosis in an AF+13 cell. Images were acquired by spinning-disk confocal microscopy at 1 min intervals and they are played back at 5 frames per second. DNA is shown green (H2B-GFP) and microtubules in red (RFP-tubulin).

revealed the presence of nuclei with four signals per chromosome (*Figure 3E–F*). These observations suggested a possible causal link between trisomy 13 and tetraploidy. Acknowledged mechanisms of tetraploidy induction include mitotic slippage (*Rieder and Maiato, 2004*), cytokinesis failure (*Normand and King, 2010*), and cell fusion (*Duelli and Lazebnik, 2003*). To determine which of these mechanisms cause tetraploidy in DLD1+13 and AF+13 cells, we performed phase contrast time-lapse microscopy and found no evidence of mitotic slippage or cell fusion. Instead, we found that both DLD1+13 and AF+13 cells failed cytokinesis (*Figure 4A–B*, *Videos 3–6*) at significantly higher rates than their diploid counterparts (*Figure 4C*). To identify the molecular mechanism that causes cytokinesis failure in cells with trisomy 13, we referred to microarray data available for DLD1+13 cells (*Upender et al., 2004*). Interestingly, located on chromosome 13q13.3 is the gene *SPG20*, which encodes for the protein Spartin, previously suggested to act as a regulator of cytokinesis (*Renvoise et al., 2010*; *Lind et al., 2011*), and shown to be overexpressed in DLD1+13 compared to DLD1 cells (*Upender et al., 2004*). None of the other mis-expressed

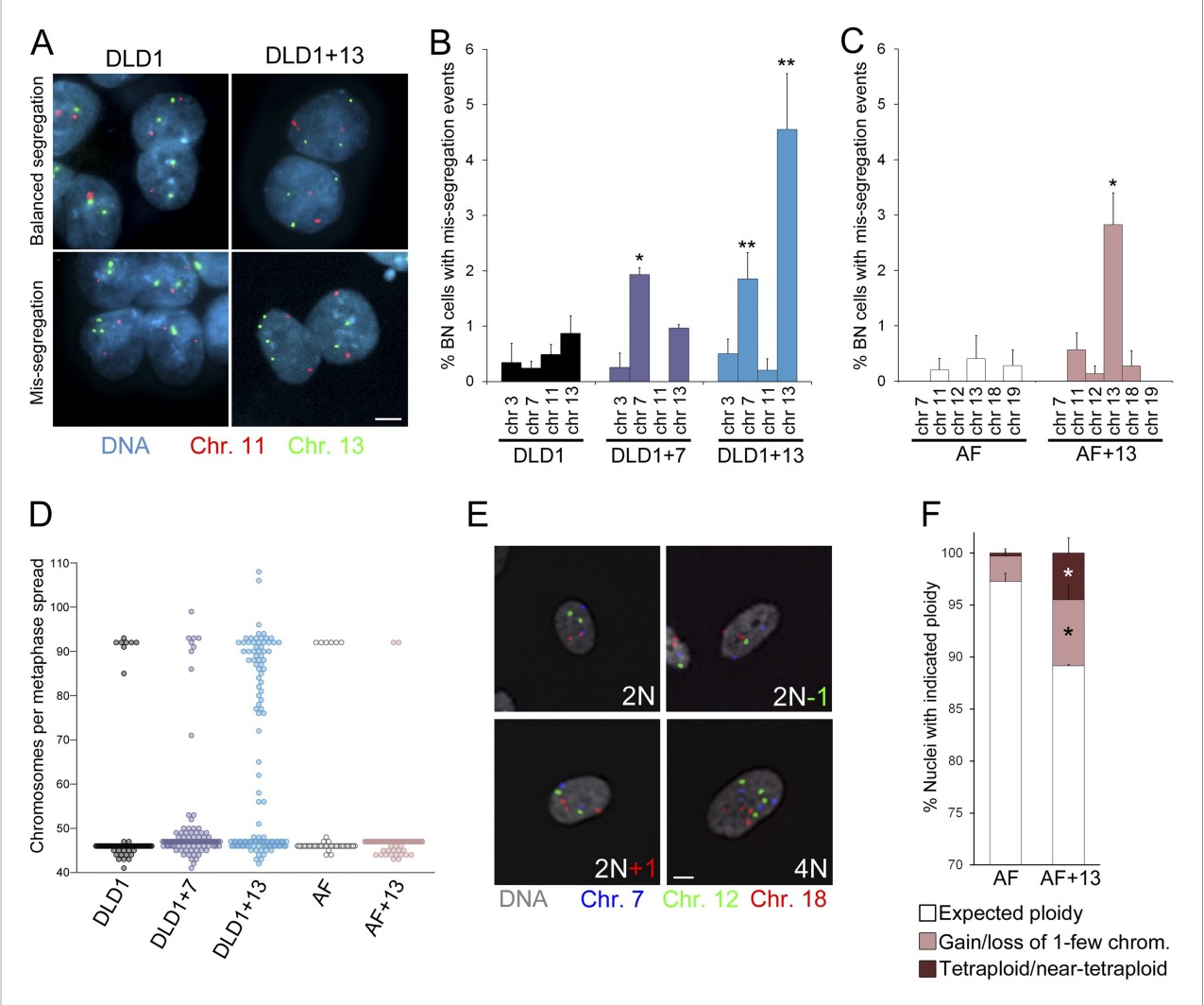

Figure 3. Increased chromosome mis-segregation rates and karyotypic heterogeneity in cells with trisomy 7 or 13. (A–C) Combination of cytokinesis-block assay and FISH staining with chromosome-specific probes shows higher chromosome mis-segregation rates in DLD1+7 and DLD1+13 compared to DLD1 cells and AF+13 compared to AF cells. (A) Examples of FISH-stained binucleate (BN) DLD1 and DLD1+13 cells. Scale bar, 5 μm. (B–C) Frequencies of BN cells displaying mis-segregation events. *Two-tailed $\chi^2$ test, p < 0.005; **two-tailed $\chi^2$ test, p < 0.0001, when compared to mis-segregation of the same chromosome in the euploid cell line. Data are presented as mean ± S.E.M. and represent the average of at least three independent experiments/samples in which a total of 460–1229 BN cells were analyzed. (D) Beeswarm plot displaying data from chromosome counts in metaphase spreads from the five cell lines. DLD1+7, DLD1+13, and AF+13 (modal chromosome number 47, shown by the high concentration of sampled points) displayed increased karyotypic heterogeneity compared to DLD1 and AF cells (modal chromosome number 46, shown by the high concentration of sampled points), respectively. In addition, DLD1+13 cells displayed a large sub-population of near-tetraploid cells (modal chromosome number 92). Chromosome counts were performed on 89–303 metaphase spreads. (E–F) FISH staining with chromosome-specific probes in interphase nuclei reveals higher rates of aneuploidy and tetraploidy in AF+13 vs AF cells. (E) FISH staining in interphase AF and AF+13 cells with probes specific for chromosomes 7 (blue), 12 (green), and 18 (red). Scale bar, 5 μm. (F) Quantification of interphase FISH data shown in (E). Cells were classified as having gained or lost 1-few chromosomes or as tetraploid (4 signals for each of the three chromosomes analyzed). The data show a larger fraction of cells with gain/loss of 1-few chromosomes in the AF+13 population compared to AF cells (*$\chi^2$ test, p < 0.0001). Additionally, AF+13 cells displayed a larger sub-population of tetraploid cells (*$\chi^2$ test, p < 0.0001). Data are presented as mean + S.E.M. and represent the average of three different samples in which a total of 2,117–2,410 cells were analyzed.

The following figure supplement is available for figure 3:

Figure supplement 1. Combined cytokinesis-block assay and FISH staining with chromosome-specific probes.

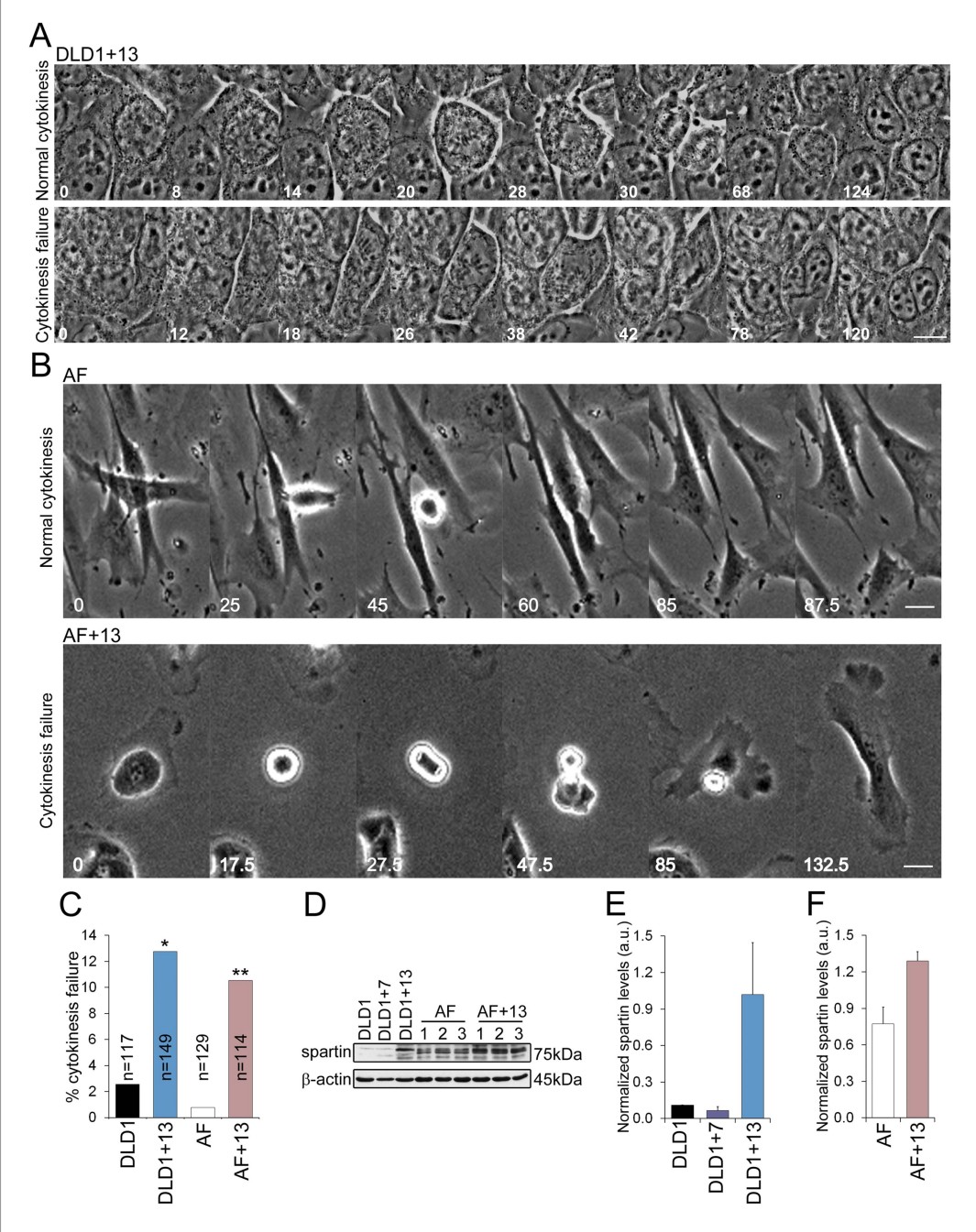

**Figure 4**. DLD1+13 and AF+13 cells overexpress SPG20 and fail cytokinesis at high rates. (**A–B**) Time-lapse microscopy indicates that cells with trisomy 13 frequently fail cytokinesis. Time stamps indicate elapsed time in minutes. Scale bars, 10 μm. (**A**) Still images from time-lapse phase contrast videos of DLD1+13 cells undergoing mitosis and completing (top row) or failing (bottom row) cytokinesis. (**B**) Still images from time-lapse phase contrast videos of AF (top row) and AF+13 (bottom row) cells undergoing mitosis and completing (AF, top row) or failing (AF+13, bottom row) cytokinesis. (**C**) Quantification of cytokinesis failure from phase-contrast time-lapse videos showing that the rates of cytokinesis failure in DLD1+13 and AF+13 cells are significantly higher than those observed in their diploid counterparts, DLD1 and AF cells (*$\chi^2$ test, p < 0.01; ** $\chi^2$ test, p < 0.001). (**D**) Western blot analysis of Spartin across DLD1 cell lines, three samples of AF and three samples of AF+13 cells. β-actin was used as a loading control. (**E**) Quantification of spartin levels (normalized to β-actin) in DLD1, DLD1+7, and DLD1+13 cells. The data reported are the average of three independent experiments and are displayed as mean + S.E.M. (**F**) Quantification

*Figure 4. continued on next page*

*Figure 4. Continued*

of spartin levels (normalized to β-actin) in AF and AF+13 cells. The data reported are the average of the three AF and AF+13 samples shown in (**E**) and are displayed as mean + S.E.M.

The following figure supplement is available for figure 4:

**Figure supplement 1**. Trisomies other than 13 are not associated with spartin overexpression.

genes (*Upender et al., 2004*) was found to have any link with cytokinesis based on published data. We confirmed SPG20 overexpression by western blot in both DLD1+13 and AF+13 cells (*Figure 4D–F*). Importantly, neither DLD1+7 (*Figure 4E*) nor other trisomic AF cells (*Figure 4—figure supplement 1*) overexpressed spartin, indicating that high levels of spartin are specifically associated with trisomy 13.

To test whether *SPG20* overexpression could explain the cytokinesis-failure phenotype, we transfected the parental cell line DLD1 with YFP-SPG20 (DLD1-YFP-SPG20; *Figure 5A*, *Videos 7–8*), and found that high levels of Spartin (*Figure 5B*) induced high rates of cytokinesis failure (*Figure 5C*). Moreover, we could rescue the cytokinesis failure phenotype in both DLD1+13 and AF+13 cells by siRNA-mediated Spartin knockdown (*Figure 5D–G*). Thus, we conclude that the aneuploidy-dependent overexpression of Spartin in DLD1+13 and AF+13 cells induces cytokinesis failure, a karyotype-dependent phenotype.

## How does spartin overexpression induce cytokinesis failure?

To determine how Spartin overexpression may lead to cytokinesis failure, we analyzed the amount and localization of Spartin in fixed DLD1 cells (*Figure 6A–B*, *Figure 6—figure supplement 1*). Spartin localized to the centrosomes throughout mitosis and to some extent along the microtubules of the mitotic spindle (*Figure 6—figure supplement 1*), and localized to the midbody during cytokinesis (*Figure 6A*), as previously described (*Lind et al., 2011*). We quantified the total intracellular amount of Spartin by measuring total Spartin fluorescence intensity in interphase cells, and found that it was

significantly higher in DLD1+13 cells compared to DLD1 cells (*Figure 6B*; see also *Figure 4D*). However, we did not observe any difference in Spartin localization between the two cell lines during mitosis (*Figure 6A*, *Figure 6—figure supplement 1*), although there was clearly a large

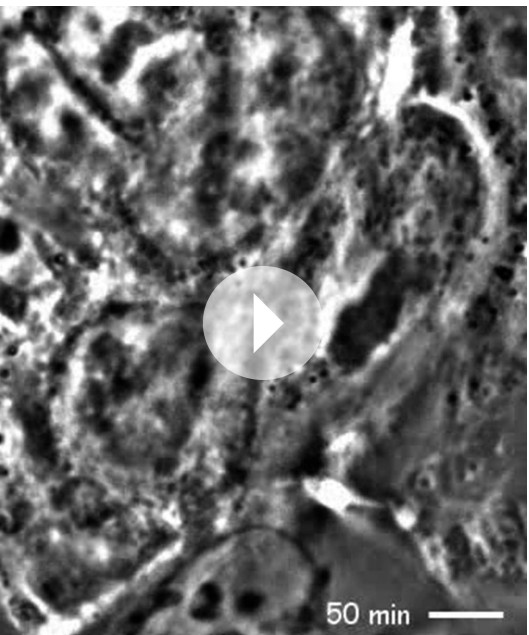

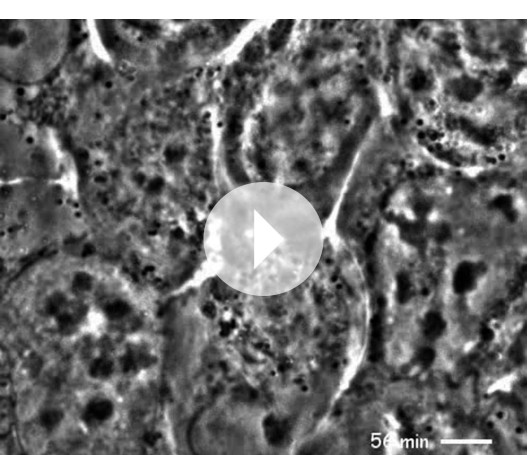

**Video 3.** Representative video showing normal cytokinesis in a DLD1+13 cell. Images were acquired by phase contrast microscopy at 2 min intervals, and they are played back at 7 frames per second. Scale bar, 5 μm.

**Video 4.** Representative video showing cytokinesis failure in a DLD1+13 cell. Images were acquired by phase contrast microscopy at 2 min intervals, and they are played back at 7 frames per second. Scale bar, 5 μm.

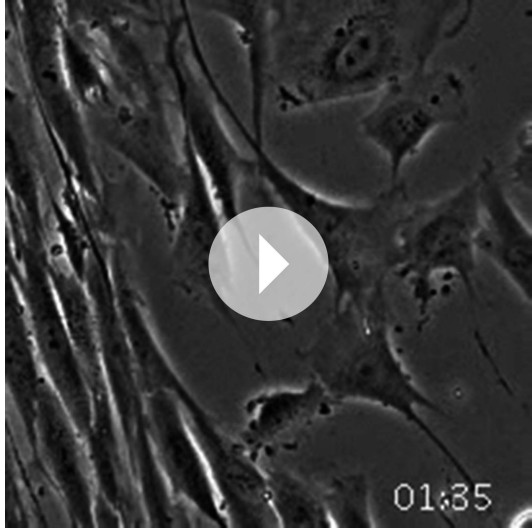
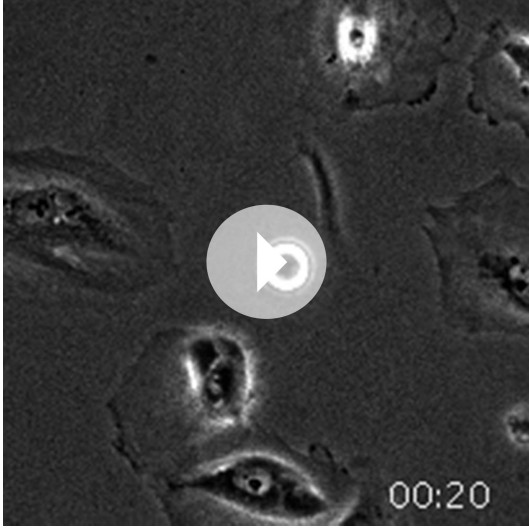

**Video 5.** Representative video showing normal cyto-kinesis in an AF cell. Images were acquired by phase contrast microscopy at 2.5 min intervals, and they are played back at 5 frames per second. Scale bar, 5 µm.

**Video 6.** Representative video showing cytokinesis failure in an AF+13 cell. Images were acquired by phase contrast microscopy at 2.5 min intervals, and they are played back at 5 frames per second. Scale bar, 5 µm.

amount of Spartin in the cytoplasm, away from the mitotic spindle, in DLD1+13, but not in the other cell lines (*Figure 6—figure supplement 1*). Thus, although Spartin overexpression induces cytokinesis failure (*Figure 5A–C*), the mechanism by which this happens is not simply mis-localization of Spartin at the midbody (*Figure 6A*).

Spartin is recruited to the midbody by binding hIST1 (*Renvoise et al., 2010*), a component of the ESCRTIII complex, which binds various proteins involved in cytokinesis, including the microtubule severing protein Spastin (*Renvoise et al., 2010*), whose depletion was shown to cause cytokinesis failure (*Bajorek et al., 2009*). Both Spartin and Spastin bind hIST1 through their MIT (Microtubule Interacting and Trafficking) domains, which show considerable structural homology (*Figure 6C*) and comparable binding affinities (Spartin, $K_d = 10.4 \pm 0.3$ µM, Spastin, $K_d = 4.6 \pm 0.1$ µM, respectively [*Renvoise et al., 2010*]). Therefore, we postulated that Spartin overexpression might act in a dominant negative manner by preventing Spastin from binding to hIST1. To test this, we analyzed Spastin localization at the midbody. Consistent with our hypothesis, DLD1+13 and AF+13 cells frequently lacked Spastin at the midbody (*Figure 6D–G*). Moreover, by knocking down Spartin we could rescue Spastin mis-localization fully in DLD1+13 (*Figure 6E*) and partially in AF+13 cells (*Figure 6G*).

In summary, we showed that overexpression of SPG20, a gene on chromosome 13 encoding the protein Spartin, can cause cytokinesis failure in cells with trisomy 13 (*Figures 4–5*). Although Spartin overexpression may cause cytokinesis failure by interfering with multiple pathways, here we provide evidence of interference with a pathway responsible for Spastin localization at the midbody (*Figure 6D–G*).

## Discussion

### Chromosomal instability in trisomic cells

We show here that cells with trisomy 7 or trisomy 13 display rates of anaphase lagging chromosomes that are significantly higher than the rates observed in euploid counterparts. Anaphase lagging chromosomes are a major source of aneuploidy in normal vertebrate cells (*Cimini et al., 2001*) and the main type of chromosome segregation defect observed in CIN cancer cells (*Thompson and Compton, 2008*; *Bakhoum et al., 2014*). Previous studies have shown that anaphase lagging chromosomes can be caused by transient spindle multipolarity in CIN cancer cells (*Ganem et al., 2009*;

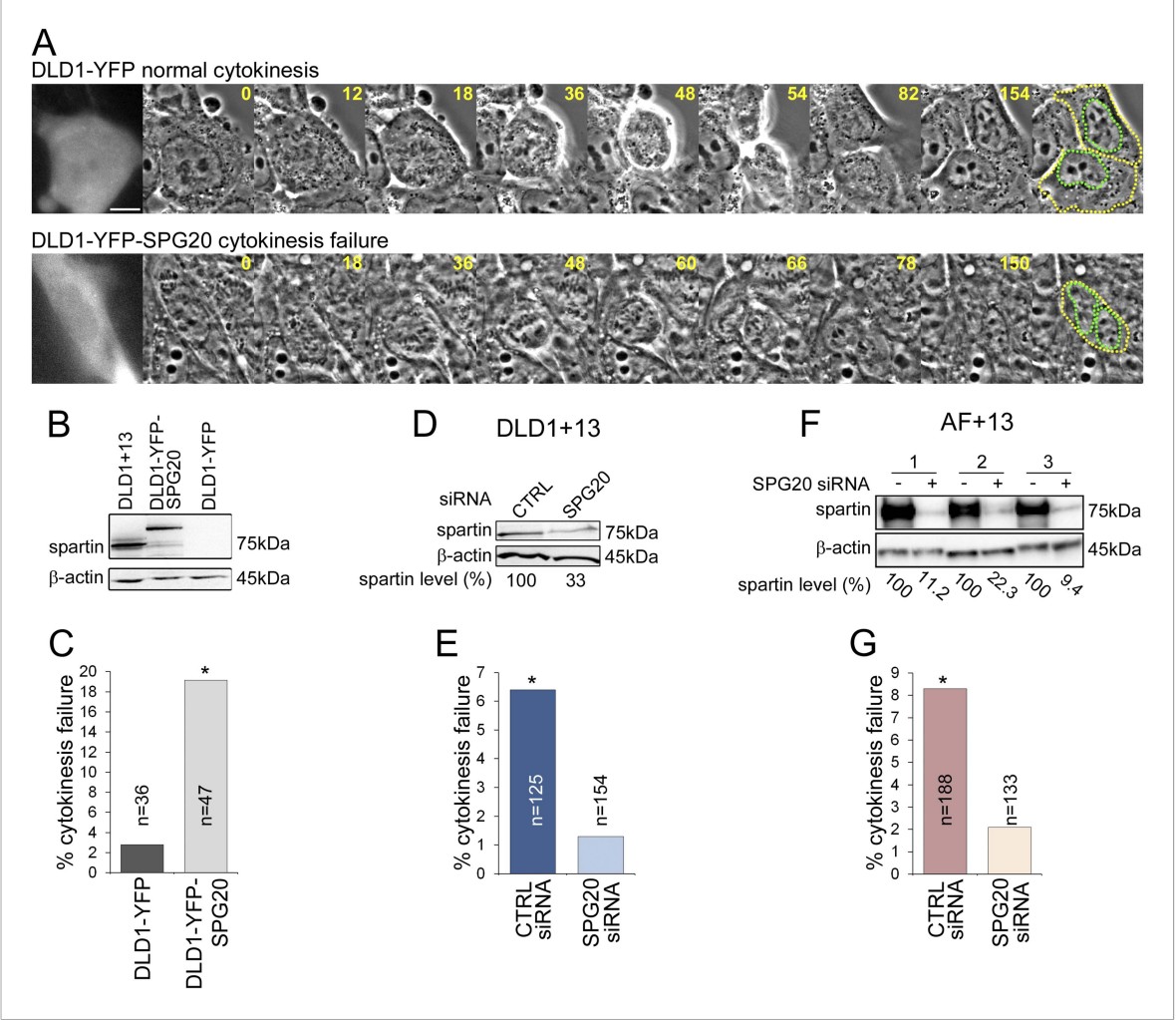

**Figure 5**. Spartin overexpression induces cytokinesis failure. (**A**) Time-lapse microscopy of DLD1 cells transiently transfected with a YFP-N1 vector (DLD1-YFP, top row) or a YFP-SPG20-N1 vector (DLD1-YFP-SPG20, bottom row). Representative still images of time-lapse videos show a DLD1-YFP cell undergoing mitosis and completing cytokinesis (top row) and a DLD1-YFP-SPG20 cell undergoing mitosis and failing cytokinesis (bottom row). YFP expression was verified by fluorescence imaging and it is shown in the first panel for each time-lapse series. A copy of the last frame was added at the end of the sequence to highlight cell (yellow) and nuclear (green) outlines. Scale bar, 10 μm. (**B**) Western blot analysis of Spartin levels in DLD1+13, DLD1-YFP, and DLD1-YFP-SPG20 cells shows that the levels of YFP-Spartin in DLD1-YFP-SPG20 cells (center lane) are much higher than the levels of Spartin in DLD1-YFP cells. WB of Spartin in DLD1+13 cells is shown for comparison. (**C**) Quantification of cytokinesis failure rates in DLD1-YFP and DLD1-YFP-SPG20 showing that overexpression of SPG20 leads to increased rates of cytokinesis failure (*$\chi^2$ test, p < 0.01 when comparing DLD1-YFP-SPG20 to DLD1-YFP cells). (**D**) Western blot analysis of Spartin levels in DLD1+13 cells treated with a SPG20 siRNA or with a control siRNA. (**E**) Reducing the levels of Spartin by SPG20 siRNA significantly reduces the rate of cytokinesis failure in DLD1+13 cells (*$\chi^2$ test, p < 0.02 when comparing cells treated with a control siRNA to cells treated with SPG20 siRNA). (**F**) Western blot analysis of Spartin levels in AF+13 cells treated with a SPG20 siRNA or with a control siRNA. (**G**) Reducing the levels of Spartin by SPG20 siRNA significantly reduces the rates of cytokinesis failure in AF+13 cells (*$\chi^2$ test, p < 0.02 when comparing cells treated with a control siRNA to cells treated with SPG20 siRNA). Average values from three independent experiments; variability between AF+13 samples was not significant.

*Silkworth et al., 2009*; *Silkworth and Cimini, 2012*). However, we can exclude transient multipolarity as a cause of anaphase lagging chromosomes in our experimental systems, given that we did not find differences in the frequencies of multipolar mitoses in DLD1+7 and DLD1+13 compared to DLD1 cells and we did not observe transient spindle multipolarity in live AF+13. This finding may seem surprising, particularly considering that we found trisomy 13 to be associated with cytokinesis failure, which is believed to result in extra centrosomes and multipolar mitoses (*Storchova and Pellman, 2004*; *Fujiwara et al., 2005*). However, this finding is in agreement with recent findings showing that

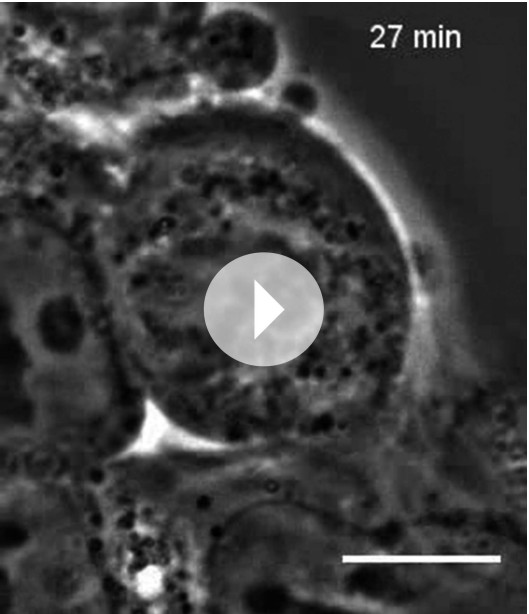

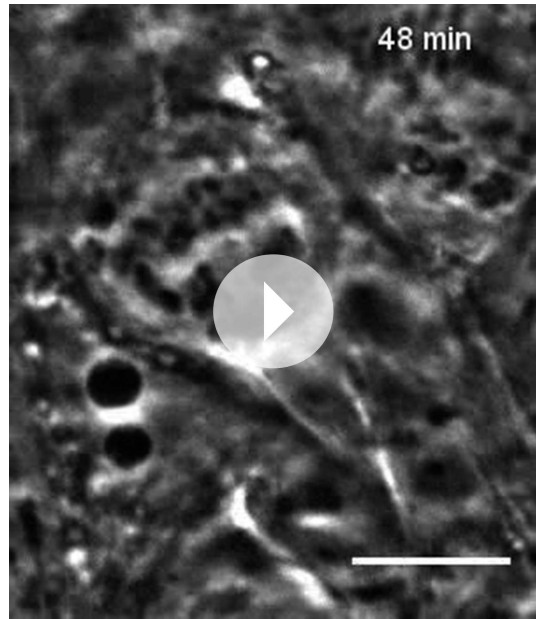

**Video 7.** Representative video showing normal cyto-kinesis in a DLD1 cell transiently transfected with a YFP vector (control). Near-simultaneous phase contrast and epifluorescence images were acquired at 4 min intervals at a single focal plane using the Nikon perfect focus function. For clarity, the fluorescent image is displayed only in the first frame, whereas the rest of the video shows only phase contrast images played back at 7 frames per second. Scale bar, 10 μm.

**Video 8.** Representative video showing cytokinesis failure in a DLD1 cell transiently transfected with a YFP-SPG20 vector (SPG20 overexpression). Near-simultaneous phase contrast and epifluorescence images were acquired at 4 min intervals at a single focal plane using the Nikon perfect focus function. For clarity, the fluorescent image is displayed only in the first frame, whereas the rest of the video shows only phase contrast images played back at 7 frames per second. Scale bar, 10 μm.

experimental inhibition of cytokinesis can produce tetraploid cells with normal centrosome number (*Godinho et al., 2014*). Our finding that aneuploidy is associated with increased rates of anaphase lagging chromosomes, known to arise from errors in mitosis (*Bakhoum et al., 2014*), but not with multipolarity or chromosome bridges, known to arise from errors in centrosome duplication or DNA metabolism (occurring prior to mitosis), suggests that compared to events occurring during other cell cycle stages, mitotic events may be more sensitive to the gene imbalance brought about by aneuploidy. Although abnormal chromosome number was reported by others as not being sufficient to cause CIN (*Lengauer et al., 1997*; *Valind et al., 2013*), such difference may be due to the different methods used to evaluate CIN. For example, whereas we examined chromosome mis-segregation in mitosis (anaphase lagging chromosomes) or at a post-mitotic interphase (BN cell analysis), Lengauer and colleagues determined the degree of aneuploidy in the overall population after 25 serial passages (*Lengauer et al., 1997*). This kind of analysis may produce a biased result because selective pressure against arising aneuploid cells may mask the ability of aneuploidy to induce chromosome mis-segregation at each cell cycle. Importantly, we were able to perform high-resolution live cell imaging of individual human amniocytes with constitutional trisomy 13 at passage number 2–3 and directly measure how frequently chromosome mis-segregation events occur. The frequency of lagging chromosomes found in mitotic cells was considerably higher compared to the frequency of chromosome mis-segregation measured by FISH analysis both in interphase and post-mitotic fixed cells. This shows how different results are generated from distinct methods and might also explain the divergence between our data and those previously reported by *Valind et al. (2013)* using interphase FISH analysis. It should also be noted that, although we do observe an increase in the rates of anaphase lagging chromosomes in trisomic cells, such rates are lower than those reported for most CIN cancer cell lines (*Thompson and Compton, 2008*; *Nicholson and Cimini, 2013*; *Bakhoum et al., 2014*), in

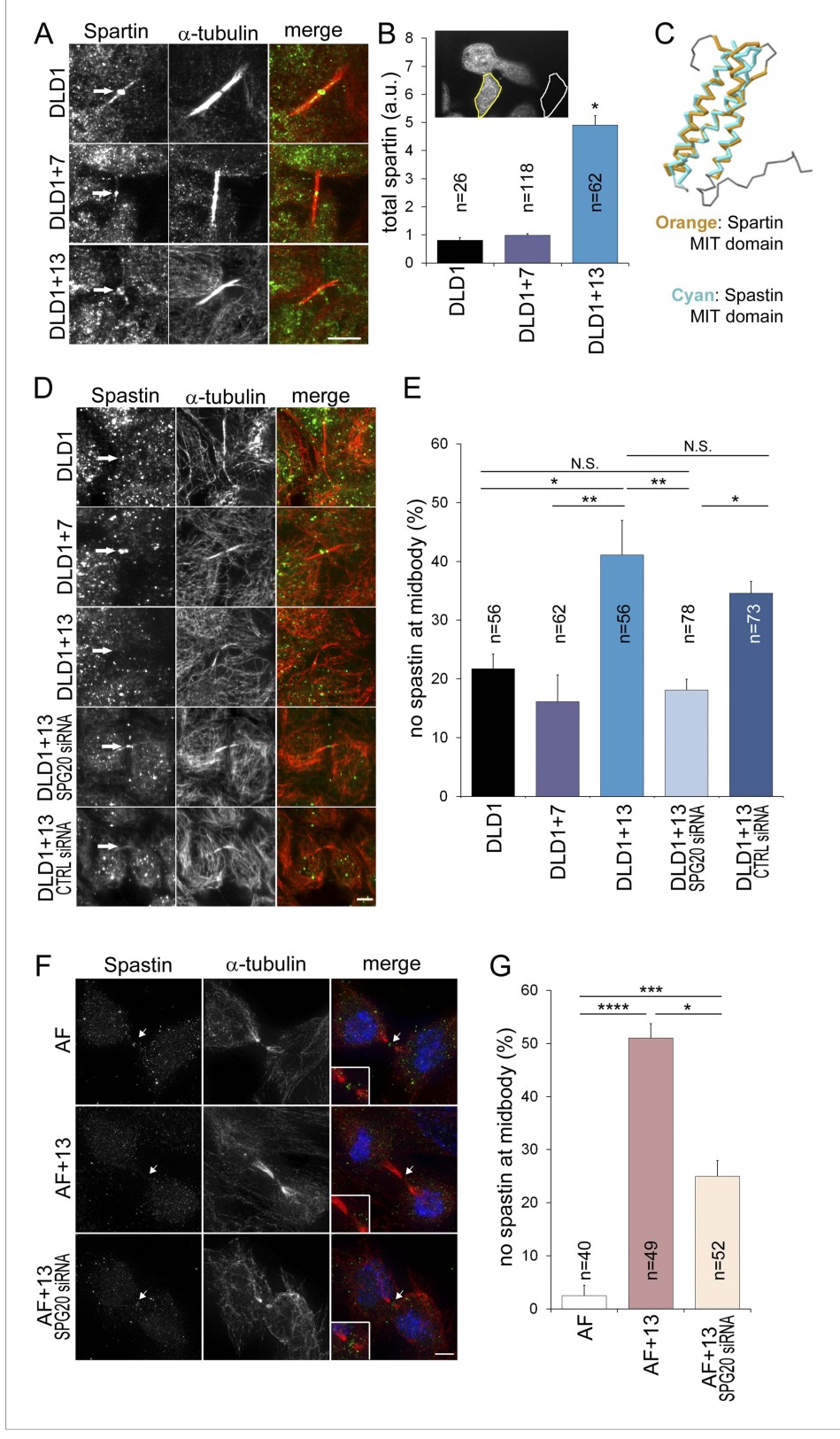

**Figure 6**. Spartin overexpression impairs Spastin localization to the midbody. (**A–B**) Spartin localization at the midbody is not affected by the high levels of intracellular Spartin in DLD1+13. (**A**) Images showing Spartin localization (arrows) at the midbody of the three DLD1 cell lines. (**B**) The image shows interphase DLD1+13 cells

*Figure 6. continued on next page*

*Figure 6. Continued*

immunostained for Spartin and the yellow and white outline indicate the regions of interest (ROI) selected for measurements of total intracellular fluorescence (yellow ROI) and background fluorescence (white ROI). The data in the graph report the total intracellular Spartin fluorescence intensity after background subtraction in randomly sampled interphase cells and are represented as mean ± S.E.M (*t-test, p < 0.0001 when comparing DLD1+13 to either DLD1 or DLD1+7). (**C**) Superimposition of the MIT domains of Spartin (orange) and Spastin (cyan) illustrates the considerable degree of structural homology between the two. PDB ID #2DL1 (*Suetake et al., 2009*) and #3EAB (*Yang et al., 2008*) for Spartin and Spastin, respectively. (**D–E**) Spastin localization at the midbody is impaired in DLD1+13 cells, and is rescued by SPG20 siRNA. (**D**) Images of midbodies of cells immunostained for Spastin (green) and microtubules (red). Arrows point at sites of Spastin localization or lack thereof. (**E**) Frequencies of cells lacking Spastin at the midbody. Higher frequencies of cells lacking Spastin at the midbody can be found in DLD1+13 cells compared to either DLD1 or DLD1+7 cells. Spastin localization could be rescued by knocking down Spartin levels in DLD1+13 cells via SPG20 siRNA. Statistical significance was calculated using a $\chi^2$ test (*p < 0.05; **p < 0.01; N.S. = not significant). Data are presented as mean ± S.E.M. and represent the average of three independent experiments. (**F–G**) Spastin localization at the midbody is impaired in AF+13 cells, and is partially rescued by SPG20 siRNA. (**F**) Images of midbodies of cells immunostained for Spastin (green) and microtubules (red). DNA is shown in blue in the merged images. Arrows point at sites of Spastin localization or lack thereof. The insets show 3× magnifications of the midbody region. (**G**) Frequencies of cells lacking Spastin at the midbody are higher in AF+13 compared to AF cells. Spastin localization was partially rescued by SPG20 siRNA-mediated Spartin knock down in AF+13 cells. Statistical significance was calculated using a $\chi^2$ test (*p < 0.05; ***p < 0.001; ****p < 0.0001). Data are presented as mean ± S.E.M. and represent the average of three independent experiments/samples. Scale bars, 5 µm.

The following figure supplement is available for figure 6:

**Figure supplement 1**. Spartin localization during mitosis.

agreement with findings by *Valind et al. (2013)*. This suggests that, although single chromosome gains may result in only modest increases in chromosome mis-segregation rates, the high degrees of aneuploidy typical of most cancer cells may have a cumulative effect and could thus explain the high rates of CIN displayed by many cancer cells (e.g., 20–75% anaphase lagging chromosomes in cells with modal chromosome number 66–78; [*Lengauer et al., 1997*; *Thompson and Compton, 2008*; *Ganem et al., 2009*; *Silkworth et al., 2009*; *Nicholson and Cimini, 2011*, *2013*]). Finally, and perhaps most importantly, the discrepancies in the conclusions reached in different studies may depend on the specific chromosomes analyzed. Indeed, we find that the degree and type of CIN elicited by aneuploidy depend on the aneuploid chromosome (see *Figure 3D*). This is an agreement with findings in aneuploid budding yeast strains, in which different aneuploidies were found to result in different rates of chromosome loss/CIN (*Sheltzer et al., 2011*; *Zhu et al., 2012*). Similarly, Valind and colleagues reported increased rates of CIN for certain chromosomes in specific aneuploid contexts (e.g., increased CIN for chromosome 17 in cells with trisomy 18), but not in others (*Valind et al., 2013*). In our study, we specifically found that high rates of mis-segregation for chromosome 7 were observed both in DLD1+7 and DLD1+13 cells, whereas high mis-segregation for chromosome 13 was observed in DLD1+13 and AF+13 cells. The observation that the trisomic chromosome displayed the highest rates of mis-segregation across the trisomies studied (*Figure 3B–C*) suggests that the aneuploid chromosomes may undergo changes that affect their mitotic behavior and segregation, such as delayed replication and/or delayed condensation timing (DRT and DCT, respectively). Indeed, aneuploidy was shown to correlate with DRT and DCT (*Grinberg-Rashi et al., 2010*) and previous studies in trisomic cells showed that one of the chromosomes of the trisomic set displays DRT (*Kost-Alimova et al., 2004*). On the other hand, the finding that chromosome 7 mis-segregated in DLD1+13, but not in AF+13 cells raises the question as to whether types and rates of mis-segregation may vary in a cell type-dependent manner and whether copy number variations in the DLD1-derived cell lines may account for chromosome 7 mis-segregation. Given that the gain of chromosome 7 and chromosome 13 is commonly found in colon cancer (*Ried et al., 2012*), a cell type-specific effect is plausible.

Although our cytokinesis-block assay data (*Figure 3B–C*) show the higher mis-segregation rates to be limited to certain chromosomes, the chromosome count data show extensive karyotypic heterogeneity in the trisomic cell populations (*Figure 3D–F*), thus suggesting that chromosome mis-segregation is more widespread than the cytokinesis-block assay reveals. One explanation is that

the limited number of cells and chromosomes analyzed in the cytokinesis-block assay might only reveal differences in mis-segregation rates when such differences are large, but other methods, such as anaphase lagging chromosomes may be a better indicator of general mis-segregation rates. Moreover, some mis-segregation events may result in cell cycle arrest or cell death. Cases of chromosome mis-segregation leading to cell death would only be accounted for when examining cells undergoing mitosis, but not interphase cells. On the other hand, chromosome mis-segregation events leading to cell cycle arrest may only be appreciated when examining interphase nuclei, but not BN or mitotic cells (*Figure 3D–F*). These considerations argue for the use of multiple assays in studies aimed at dissecting the link between aneuploidy and CIN (*Nicholson and Cimini, 2015*). We would also like to point out that chromosome mis-segregation events causing cell death or cell cycle arrest under tissue culture conditions, may not do so in the context of the tumor environment, suggesting that low level aneuploidy could be enough to drive CIN in cancer. Indeed, our data show that the complex aneuploidies observed in cancer cells are not a requirement for increased rates of lagging chromosomes and that at least some low grade and constitutional aneuploidies are sufficient to induce such an effect. This would be in agreement with previous observations showing that haploid budding yeast strains carrying disomies displayed increased genomic instability, and strains with different degrees of aneuploidy displayed variable degrees of chromosomal instability (*Sheltzer et al., 2011*; *Zhu et al., 2012*). Similarly, previous reports have shown that lymphocytes of congenitally trisomic individuals displayed aneuploidies for chromosomes other than the congenitally trisomic ones (*Reish et al., 2006*, *2011*). Finally, random aneuploidy and senescent phenotypes were recently reported in aneuploid amniocytes (*Biron-Shental et al., 2015*). Nonetheless, we do not exclude the possibility that certain aneuploidies may not be sufficient to induce chromosome mis-segregation.

## Aneuploidy confers karyotype-dependent phenotypes

Our finding that trisomy 13 caused a specific cytokinesis failure phenotype (*Figures 4–5*) clearly indicates that different karyotypes can be associated with distinct phenotypic changes. It is important to note that, although we observed cytokinesis failure in AF+13, the frequency of tetraploid metaphase spreads in these cells was very low, as opposed to the DLD1+13 cells (*Figure 3D*). This may be due to activation of a post-mitotic tetraploidy checkpoint in the AF+13 cells, but not in DLD1+13. Indeed, previous studies have shown a cell cycle arrest following cleavage failure in untransformed human cells (*Andreassen et al., 2001*; *Krzywicka-Racka and Sluder, 2011*), whereas transformed cells continue cycling (*Duelli et al., 2007*; *Panopoulos et al., 2014*), but the triggering of a p53-dependent arrest in tetraploid cells still remains a matter of debate (*Andreassen et al., 2001*; *Stukenberg, 2004*; *Uetake and Sluder, 2004*; *Fujiwara et al., 2005*; *Wong and Stearns, 2005*).

The observation that overexpression of a gene mapping on chromosome 13 is specifically linked to the cytokinesis failure phenotype in DLD1+13 and AF+13 cells demonstrates that there is a direct causal relationship between aneuploidy, overexpression of genes on the aneuploid chromosome, and phenotypic changes caused by the consequent proteomic imbalance. These findings support previous studies showing that aneuploidy directly affects transcript and protein levels in various systems in a karyotype-dependent manner (*Pollack et al., 2002*; *Upender et al., 2004*; *Gao et al., 2007*; *Pavelka et al., 2010b*; *Ried et al., 2012*; *Stingele et al., 2012*; *Gemoll et al., 2013*). Previous studies in budding yeast also showed that aneuploid strains displayed karyotype-specific phenotypic variations that conferred resistance to a variety of drugs (*Pavelka et al., 2010b*). However, such phenotypic variations in aneuploid yeast strains were only revealed when cells were grown under specific selective conditions (*Pavelka et al., 2010b*), whereas we show here that such karyotype-dependent phenotypic changes can be intrinsic to the aneuploid cells. Although some studies have shown that aneuploidy can have overall deleterious effects on cell fitness (*Torres et al., 2007*; *Williams et al., 2008*), we provide strong evidence that specific aneuploidies can also induce specific phenotypes, which could, under certain conditions, provide a selective advantage. This particular concept was recently exploited to demonstrate that in fungi, certain drug treatments can lead to the evolution of populations with defined aneuploid karyotypes (*Chen et al., 2015*). In the particular case observed in our study, one could also envision how the increase in tolerance to aneuploidy that tetraploidy was recently shown to confer (*Dewhurst et al., 2014*) could enable the evolution of specific aneuploid karyotypes that may allow cells to overcome the detrimental impact of aneuploidy on cellular fitness (*Gordon et al., 2012*). Karyotype-specific phenotypic changes such as those observed in our study can also explain the recurrent aneuploidies that are found in tumors from certain anatomical sites (e.g., gain of chromosome 13 and loss of chromosome 18 in colon cancer

[*Ried et al., 1996*, *2012*; *Nicholson and Cimini, 2013*]). Indeed, aneuploidies for certain chromosomes may result in phenotypes that confer a selective advantage at a certain site (e.g., breast), but not at a different one (e.g., colon). And this could explain why, despite the high degrees of aneuploidy and the extensive karyotypic heterogeneity, the distribution of aneuploidies in different cancers is not completely random (*Nicholson and Cimini, 2011*; *Ried et al., 2012*).

## Materials and methods

### Cell lines and culture conditions

The DLD1 cell line was obtained from American Type Culture Collection (ATCC, BA, USA), DLD1+7 and DLD1+13 cell lines were created previously by microcell-mediated chromosome transfer as described in (*Upender et al., 2004*), and DLD1+13 cells were sub-cloned for this study as described in the results section. All cell lines were maintained in RPMI 1640 (ATCC, BA, USA) supplemented with 10% FBS (Gibco, Life Technologies, CA, USA), penicillin, streptomycin, and amphotericin B (antimycotic). Passage 1–3 fibroblast cultures were established from surplus amniocentesis samples used in pre-natal diagnosis. Three cases of constitutional trisomy 13 and three diploid controls were used in our study (*Table 1*). The study acknowledged the ethics guidelines under national rules and according to the principles of the Declaration of Helsinki, and was approved by the Ethics Committee of Hospital de S. João-Porto (dispatch 14 November 2012). Informed consent forms with detailed information were provided to all patients. The study did not imply collection of extra material from the healthy female donors (only surplus cells/tissues were used); the study did not bring any direct benefits to the volunteers; there were no risks or costs for the volunteers; there was no access to patient clinical data (samples were obtained in anonymous form from the Hospital Genetics Department); participation was volunteer and free to be interrupted at any moment; there are no ethical impacts predicted; there will be no commercial interests. Amniotic fibroblasts were grown in EMEM (Lonza, Bazel, Switzerland) supplemented with 15% FBS, 2.5 mM glutamine and 1× antibiotic-antimycotic solution (all from Gibco, Life Technologies, CA, USA). All cells were kept in a humidified incubator at 37°C with 5% $CO_2$.

### Immunostaining

For immunostaining, DLD1 cells were grown on sterilized glass coverslips inside 35 mm Petri dishes, whereas AF cells were grown on sterilized glass coverslips coated with fibronectin (Sigma Aldrich, MO, USA). For analysis of anaphase lagging chromosomes in the DLD1 lines, cells were fixed in freshly prepared 4% paraformaldehyde in PHEM (60 mM Pipes, 25 mM HEPES, 10 mM EGTA, 2 mM $MgSO_4$, pH 7.0) for 20 min at room temperature and then permeabilized for 10 min at room temperature in PHEM buffer containing 0.5% Triton-X 100. Following fixation and permeabilization, cells were washed with PBS 3 times and then blocked with 10% boiled goat serum (BGS) for 1 hr at room temperature. Cells were then incubated at 4°C overnight with primary antibodies diluted in 5% BGS. Cells were washed in PBS-T (PBS with 0.05% Tween 20) 3 times, and incubated at room temperature for 45 min with secondary antibodies diluted in 5% BGS. Cells were finally washed, stained with DAPI for 5 min, and coverslips were mounted on microscope slides in an antifade solution containing 90% glycerol and 0.5% N-propyl gallate. For analysis of proteins at the midbody, cells were fixed in ice cold methanol for 4 min, washed with PBS 3 times, and blocked in 10% BGS with 0.5% Triton-X for 1 hr. The rest of the procedure was the same as described above, except that primary antibodies were diluted in 1% BGS. Primary antibodies were diluted as follows: ACA (human anti-centromere protein, Antibodies Inc., CA, USA), 1:100; mouse anti-α-tubulin (DM1A, Sigma Aldrich, MO, USA), 1:500; rabbit anti-SPG20 (Protein Tech Group Inc., IL, USA), 1:300; mouse anti-Spastin (SP 3G11/1, Abcam, Cambridge, UK), 1:150. Secondary antibodies were diluted as follows: Rhodamine Red-X goat anti-mouse (Jackson ImmunoResearch Laboratories, Inc., PA, USA), 1:100; Rhodamine Red-X goat anti-rabbit (Jackson ImmunoResearch Laboratories, Inc., PA, USA), 1:100; Alexa 488 goat anti-human (Molecular Probes, Life Technologies, CA, USA), 1:200; Alexa 488 goat anti-mouse (Molecular Probes, Life Technologies, CA, USA), 1:200.

### Metaphase spreads

Cell cultures were incubated in 50 ng/ml Colcemid (Karyomax, Invitrogen) at 37°C for 3–4 hr to enrich in mitotically arrested cells. The cells were then collected by trypsinization and centrifuged at 800 rpm for 8 min. Hypotonic solution (0.075M KCl) was added drop-wise to the cell pellet and incubated for 10 min at room temperature. Cells were fixed with an ice-cold 3:1 methanol:acetic acid solution for

5 min and then centrifuged at 800 rpm for 8 min. This last step was repeated two more times and fixed cells were finally dropped on microscope slides.

## Cytokinesis-block assay and fluorescence in situ hybridization (FISH)

For FISH on binucleate cells, amniocytes were grown in superfrost ultra plus slides (Thermo Scientific, Life Technologies, CA, USA), whereas DLD1 cells were grown on coverslips. For the cytokinesis-block assay, cells were treated with 3 µg/ml dihydrocytochalasin B (Sigma–Aldrich, MO, USA) for 24 hr before fixation, and the experiment was repeated at least three independent times. Prior to being processed for FISH staining, cytokinesis-blocked cells were fixed according to previously published protocols: a standard FISH protocol (*Nicholson and Duesberg, 2009*) was used for all cells; an alternative (3-D FISH) protocol (*Cremer et al., 2008*) was used in some experiments with DLD1, DLD1+7, and DLD1+13 cells. Bacterial artificial chromosome (BAC) contigs using three to six BAC sequences specific to each region were made for the following four probes: *CDX2* on chromosome 13q12, *MET* on chromosome 7q31, *CHEK1* on chromosome 11q24, and *TERC* on chromosome 3q26. The BAC clone contigs were labeled by nick translation with Spectrum Orange-dUTP (Abbott Laboratories; IL, USA) for CDX2, Dy-505-dUTP (Dyomics; Jena, Germany) for MET, Spectrum Orange-dUTP (Abbott Laboratories; IL, USA) for *CHEK1*, and Dy-505-dUTP (Dyomics; Jena, Germany) for *TERC*. Dual color human whole chromosome paint probes were generated in-house using PCR labeling techniques. Chromosome 7 was labeled with Spectrum Orange (Abbott Laboratories; Chicago, IL) and chromosome 13 was labeled with Dy505 (Dyomics; Jena, Germany). Additionally, commercial locus-specific FISH probes for chromosomes 7, 11, 19 (red 5-ROX dUTP) and 12, 13, 18 (green 5-fluorescein dUTP) were also used (Empire Genomics; Buffalo, NY, USA). The probe mixtures were co-denatured with the coverslips at 72°C for 5 min before being placed in a moist chamber at 37°C for two nights. After two nights, the coverslips were washed in 2XSSC for 5 min and then mounted on microscope slides with mounting media (Vectashield; CA, USA) and DAPI.

FISH on DLD1 metaphase spreads was performed with the commercial FISH probes listed above. Additionally, centromeric FISH probe against chromosome 7 (FITC) (Cytocell, Cambridge, UK) was used in DLD1+7 cells to confirm the presence of centromeric DNA. FISH on metaphase spreads of AF and AF+13 was performed with the XA 13/21 probe (MetaSystems, Germany) according to the manufacturer's instructions. Interphase FISH was performed with the Vysis centromeric probes CEP7 Spectrum Aqua, CEP12 Spectrum Green, and CEP18 Spectrum Orange (Abbott Molecular, IL, USA) according to the manufacturer's instructions. All the FISH-stained samples were analyzed blindly.

## SPG20 transfection and SPG20 siRNA

DLD1 cells, grown on sterilized coverslips inside 35 mm Petri dishes, were transiently transfected with either P-EYFP-N1 or P-EYFP-N1-SPG20 (a kind gift from J Bakowska) using Fugene HD (Roche, Basel, Switzerland) according to the manufacturer's protocol. For SPG20 knockdown, cells grown in glass-bottom 35 mm u-dishes (Ibidi GmbH, Germany) or on sterilized coverslips inside 35 mm Petri dishes, were transfected with Silencer Select siRNA specific to SPG20 (S23057, Ambion, Life Technologies, CA, USA) using Oligofectamine according to the manufacturer's instructions (Invitrogen, Life Technologies, CA, USA). Silencer Select Negative control siRNA (Ambion, Life Technologies, CA, USA) was used as a negative control and was also transfected into cells with Oligofectamine (Invitrogen, Life Technologies, CA, USA). Cells were observed 48–72 hr after transfection.

## Western blot analysis

Whole-cell extracts were separated by SDS-PAGE and transferred to PVDF membrane. Membranes were blocked 1 hr at room temperature with 5% milk in tris-buffered saline and then incubated over night with primary antibodies at 4°C. Antibodies were diluted as follows: rabbit anti-Spartin (Protein Tech Group Inc., IL, USA), 1:1000; rabbit anti-β-actin (Abcam, Cambridge, UK), 1:500. Blots were detected using goat anti-rabbit secondary antibodies conjugated to horse radish peroxidase and visualized with SuperSignal West Femto (Thermo Scientific, Life Technologies, CA, USA). A GS-800 calibrated densitometer, or alternatively a ChemiDoc XRS system, was used for quantitative analysis of protein levels with the help of ImageLab 4.1 software (BioRad, CA, USA).

## Confocal microscopy of immunostained cells and image analysis

Immunofluorescently labeled DLD1 cells were imaged with a swept field confocal system (Prairie Technologies, WI, USA) on a Nikon Eclipse TE2000-U inverted microscope (Nikon Instruments Inc.,

NY, USA) equipped with a 100×/1.4 NA Plan-Apochromatic objective and an automated ProScan stage (Prior Scientific, Cambridge, UK). The confocal head was accessorized with multiband pass filter set for illumination at 405, 488, 561, and 640 nm, and illumination was obtained through an Agilent monolithic laser combiner (MLC400) controlled by a four channel acousto-optic tunable filter. Digital images were acquired with a HQ2 CCD camera (Photometrics, AZ, USA). Acquisition time, Z-axis position, laser line power, and confocal system were all controlled by NIS Elements AR software (Nikon Instruments Inc., NY, USA) on a PC computer (Dell, TX, USA). Both anaphase lagging chromosomes and Spartin localization were analyzed by acquiring Z-sections of cells at 0.6 μm steps. The frequencies of anaphase lagging chromosomes were determined from 3 independent experiments performed in duplicate. Spartin localization at the midbody was determined from three independent experiments. Image acquisition of Spastin immunostaining in AFs was carried out on a Zeiss AxioImager Z1 equipped with an Axiocam MR and using a Plan-Apochromat 63×/1.4 NA objective. Images of telophase cells from three independent experiments and/or samples were acquired as Z-stacks with 0.3 μM steps and scored for the presence or absence of Spastin staining at the midbody. All data were analyzed blindly.

## Microscopy and image analysis of FISH-stained binucleate cells

FISH samples were viewed and imaged either with a Leica DM-RXA fluorescence microscope (Leica; Wetzlar, Germany) or with a swept field confocal system (Prairie Technologies, WI, USA) on a Nikon Eclipse TE2000-U inverted microscope (Nikon Instruments Inc., NY, USA). The Leica DM-RXA fluorescence microscope was equipped with custom optical filters and a 63×/1.3 NA objective. The Leica CW 4000 FISH software was used to acquire multifocal images for each fluorescence channel. 15 to 25 images were taken in areas of optimal cell density with minimal cellular clumps and overlapping cells. The Nikon Eclipse TE2000-U inverted microscope was equipped with a 100×/1.4 NA Plan-Apochromatic objective and an automated ProScan stage (Prior Scientific, Cambridge, UK). The confocal head was accessorized with a multiband pass filter set for illumination at 405, 488, 561, and 640 nm and illumination was obtained through an Agilent monolithic laser combiner (MLC400) controlled by a four channel acousto-optic tunable filter. Digital images were acquired with a HQ2 CCD camera (Photometrics, AZ, USA). Exposure time, Z-axis position, laser line power, and confocal system were all controlled by NIS Elements AR software (Nikon Instruments Inc., NY, USA) on a PC computer (Dell, TX, USA). FISH-stained samples were analyzed blindly and only cases with a total even number of signals were included in the analysis.

## Phase contrast live cell imaging

DLD1, DLD1+7, and DLD1+13 cells were grown on sterilized coverslips inside 35 mm Petri dishes. Coverslips at 60–70% confluency were mounted in Rose chambers filled with L-15 medium supplemented with 4.5 g/l glucose. Images were acquired on a Nikon Eclipse Ti inverted microscope (Nikon Instruments Inc., NY, USA) equipped with phase-contrast transillumination, transmitted light shutter, ProScan automated stage (Prior Scientific, Cambridge, UK), and a HQ2 CCD camera (Photometrics, AZ, USA). Cells were maintained at ~36°C using an air stream stage incubator (Nevtek, MA, USA). For analysis of cytokinesis in untransfected or siRNA-transfected cells, 10–15 different fields of cells were imaged every 2 min for 6–8 hr using a 60×/1.4 NA Plan-Apochromatic phase contrast objective controlled by Nikon Perfect Focus (Nikon Instruments Inc., NY, USA). For P-EYFP-N1 and P-EYFP-N1-SPG20 transfection experiments, 10–15 different fields of cells were imaged by fluorescence and phase contrast using a 20× or 60× objective every 4 min for 8 hr. The time-lapse videos were analyzed using NIS Elements AR (Nikon Instruments Inc., NY, USA) software on a PC computer (Dell, TX, USA).

Amniocytes were grown in glass-bottom 35 mm u-dishes (Ibidi GmbH, Germany) coated with fibronectin and filled with phenol red-free EMEM complete medium. Images were acquired on a Zeiss Axiovert 200M inverted microscope (Carl Zeiss, Germany) equipped with a CoolSnap camera (Roper, FL, USA), XY motorized stage and NanoPiezo Z stage, under controlled temperature, atmosphere and humidity. 20–25 neighbor fields were imaged every 2.5 min for 1–2 days using a 20×/0.3 NA A-Plan objective. Grids of neighboring fields were generated using the plugin Stitch Grid (Stephan Preibisch) from open source Fiji/Image J (http://rsb.info.nih.gov/ij/).

## Spinning-disk confocal live cell imaging

Amniotic cells, glass-bottom 35 mm u-dishes (Ibidi GmbH, Germany) coated with fibronectin, were co-transfected with H2B-GFP and pmRFP-tubulin expression plasmids (Addgene, MA, USA) using Lipofectamine 3000 transfection reagent and according to the manufacturer's instructions. Live cell imaging was performed 48 hr following transfection under a spinning-disk confocal system Andor Revolution XD (Andor Technology, UK) coupled to an Olympus IX81 inverted microscope (Olympus, UK) equipped with an electron-multiplying CCD iXonEM Camera and a Yokogawa CSU-22 unit based on an Olympus IX81 inverted microscope. Two laser lines at 488 and 561 nm were used for the excitation of GFP and pmRFP and the system was driven by IQ software (Andor Technology, UK). Z-stacks (0.8–1.0 µm) covering the entire volume of the mitotic cells were collected every 1.5 min with a PLANAPO 60×/1.4 NA objective. ImageJ was used to process all the videos.

## Acknowledgements

We thank members of the Cimini, Ried, and Logarinho labs for helpful dialogue and critical comments. We particularly thank Bin He for technical assistance with *Figure 3D* and Yue Hu for advice on statistical analysis. We also thank Joanna Bakowska (Loyola University, Chicago) for generously providing the YFP-SPG20 plasmid. Work in the Cimini lab was supported by NSF grant MCB-0842551 and HFSP grant RGY0069/2010. Work in the Logarinho lab is supported by Programa Operacional Regional do Norte (ON.2) grant NORTE-07-0124-FEDER-000003. JM is the recipient of a PhD fellowship from Fundação para a Ciência e a Tecnologia (FCT) of Portugal (SFRH/BD/74002/2010).

## Additional information

### Funding

| Funder | Grant reference | Author |
| --- | --- | --- |
| National Science Foundation (NSF) | MCB-0842551 | Daniela Cimini |
| Human Frontier Science Program (HFSP) | RGY0069/2010 | Daniela Cimini |
| Programa Operacional Regional do Norte | NORTE-07-0124-FEDER-000003 | Elsa Logarinho |
| Fundação para a Ciência e a Tecnologia | SFRH/BD/74002/2010 | Joana C Macedo |

The funders had no role in study design, data collection and interpretation, or the decision to submit the work for publication.

### Author contributions

JMN, Conception and design, Acquisition of data, Analysis and interpretation of data, Drafting or revising the article; JCM, AJM, DW, JC, Acquisition of data, Analysis and interpretation of data, Drafting or revising the article; VL, SD, Optimized essential protocols for experiments with amniocytes, Final approval of the manuscript, Acquisition of data, Drafting or revising the article, Contributed unpublished essential data or reagents; AMG, Acquisition of data, Drafting or revising the article; TR, Conception and design, Drafting or revising the article, Contributed unpublished essential data or reagents; EL, DC, Conception and design, Analysis and interpretation of data, Drafting or revising the article, Contributed unpublished essential data or reagents

### Author ORCIDs

Sofia Dória, http://orcid.org/0000-0001-9225-9076

### Ethics

Human subjects: The study acknowledged the ethics guidelines under national rules and accordingly to the principles of the Declaration of Helsinki, and was approved by the Ethics Committee of Hospital de S. João-Porto (dispatch 14 November 2012) (approval number 237/2012). Informed consent forms with detailed information were provided to all patients. The study did not imply collection of extra material from the healthy donor females (only surplus cells/tissues were used); the study didn't bring any direct benefits to the volunteers; there were no risks or costs for the

volunteers; there was no access to patient clinical data (samples were obtained in anonymous form from the Hospital Genetics Department); participation was volunteer and free to be interrupted at any moment; there are no ethical impacts predicted; there will be no commercial interests.

## Additional files

### Major datasets

The following previously published datasets were used:

| Author(s) | Year | Dataset title | Dataset ID and/or URL | Database, license, and accessibility information |
|---|---|---|---|---|
| Suetake T, Hayashi F, Yokoyama S | 2009 | Solution structure of the MIT domain from human Spartin | http://www.rcsb.org/pdb/explore/explore.do?structureId=2DL1 | Publicly available at RCSB Protein Data Bank (2DL1). |
| Yang D, Rismanchi N, Renvoise B, Lippincott-Schwartz J, Blackstone C, Hurley JH | 2008 | Crystal structure of Spastin MIT in complex with ESCRT III | http://www.rcsb.org/pdb/explore/explore.do?structureId=3EAB | Publicly available at RCSB Protein Data Bank (3EAB). |

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
