## [Decision Letter]

Thank you for sending your work entitled “Aneuploidy causes karyotype-dependent phenotypes in human cells” for consideration at *eLife*. Your article has been favorably evaluated by Stylianos Antonarakis (Senior editor), a Reviewing editor, and two reviewers.

The Reviewing editor and the reviewers discussed their comments before we reached this decision, and the Reviewing editor has assembled the following comments to help you prepare a revised submission.

While both referees found your results interesting, they raised several concerns that are set out below. One referee commented that your mechanistic analysis of Spartin is interesting but based on two evidentiary leaps. First, that transient transfection of Spartin mimics the trisomic state, and second, that knockdown of Spartin mimics the disomic state. They suggested that you could strengthen this evidence using more specific techniques (lentiviral infection to introduce one copy of Spartin into the genome; CRISPR to remove one allele). These would undoubtedly strengthen your case but are substantial undertakings; therefore, in revising your study you should characterise your Spartin over expression and knock-down cell lines more carefully and discuss the potential caveats of your approaches.

Major points:

1) The findings described here are at odds with the findings by [72]. This study reported minimal chromosomal instability among aneuploid cells with eight different karyotypes. Only two different karyotypes are examined in this paper. This raises the question of the extent to which these results can be generalised.

2) Two phenotypes of aneuploid cells are described in this manuscript: cytokinesis failure and the generation of lagging chromosomes. Are these two phenotypes related or distinct? Tetraploid cells are known to be chromosomally-unstable (e.g., Fujiwara, 2005). Are the lagging chromosomes in Figure 2 occurring specifically in tetraploid cells? This could be tested by sorting a 2N population from each cell line and looking for laggards. FiFISH staining of chr. 7, 12 and 18 on interphase nuclei. How often are multipolar mitoses (which would be expected with the high frequency of tetraploidy) observed?

3) How was the tetraploidy/near tetraploidy determined? In Figure 3, there appear to be more tetraploid wild-type AF cells than tetraploid AF +13 cells. However, in Figure 3, the opposite is shown. Figure 3 shows that at least 5 % of AF+13 cells are tetraploid/near tetraploid without explaining how were these number obtained. In the text it appears that this is extrapolation from the FISH staining of chr. 7, 12 and 18 on interphase nuclei. Identification of four signals per chromosome does not provide sufficient evidence of tetraploidy.

4) How many cells from the population are really trisomic? In the original publication ([69], Cancer Research), it was shown that only approximately 60 % of the cells DLD1 + 13 contain three copies of chromosome 13 (20% disomic, rest tetrasomic). In particular in the light of the main finding of the manuscript that aneuploidy facilitates further chromosomal instability it would be very useful to know the true karyotypic variability of the analyzed cell lines (in particular when specifically the trisomic chromosome should be affected). The authors should at least provide percentage of trisomic cells within the population and show for example by multicolor FISH or by SNP arrays that no other recurrent aneuploidy occurred during the passaging. Also, providing the passage number would be useful. In the Materials and methods it is described that three cases of constitutional trisomy 13 and three diploid controls were used in the study. Is the result presented in figures as AF and AF+13 an average of these three different cell populations, or is it one example?

5) The overexpression levels of protein Spartin should be quantified from three independent experiments. Further investigation into the importance of Spartin overexpression on cytokinesis failure is important as Figure 4 demonstrates that Spartin is substantially over-expressed in DLD+13. Thus, Spartin may be upregulated in part as a general consequence of aneuploidy, and not simply because of its increased copy number. Alternatively, is the region 13q.13.3 amplified in DLD1+13?

6) The link to spastin, although intriguing, is not sufficiently substantiated to explain the observed phenotypes. How does the spartin/spastin pathway explain the increased rates of chromosome missegregation? An experiment showing that the overexpression of SPG20 in DLD1 leads to loss of spastin in midbody would be useful.

In addition to these points the referees suggested that some clarification of the text will be required to address some issues as follows:

The authors' conclude that aneuploidy induces CIN. However, the cytokinesis block experiments in Figure 3 actually argue against this statement. The authors report that only 1 out of 11 euploid chromosomes display frequent missegregation, while 3 out of 3 aneuploid chromosomes display frequent missegregation. This suggests that aneuploidy does not induce a general state of CIN, but instead aneuploid cells have particular trouble segregating aneuploid chromosomes. This discovery is certainly very interesting and worthy of further exploration. However, it undercuts the conclusion that aneuploidy causes CIN.

The authors claim that: “…addition to previously described mechanisms, [these findings] might explain the significantly higher rates of anaphase…” (in the Discussion). This implies that they found a new mechanism. However, no new mechanism of chromosome segregation errors is proposed, in particular if they state that there are no multipolar mitoses.

[Editors' note: further revisions were requested prior to acceptance, as described below.]

Thank you for resubmitting your work entitled “Aneuploidy causes chromosome mis-segregation and karyotype-dependent phenotypes in human cells” for further consideration at *eLife*. Your revised article has been favorably evaluated by Stylianos Antonarakis (Senior editor), a member of the Board of Reviewing Editors, and two reviewers. Both referees agree that the manuscript has been significantly improved but one reviewer is concerned by the generality of the conclusions that you can draw from your study. Thus, in your final, revised manuscript it will be necessary to tone down your conclusions and mention the caveats that the reviewer identifies.

*Reviewer #2*:

The authors addressed a majority of the main concerns and the manuscript has significantly improved. However, there remain some issues that are difficult to reconcile and that at this point prevent me from recommendation for publication:

1) Somehow it is difficult to accept the general conclusion—“Aneuploidy causes chromosome mis-segregation”—when it is made on the basis of analysis of two different trisomies in only one cancerous and p53 deficient cell line (AF cells only with trisomy 13 were analyzed). Thus, this sweeping conclusion with potentially large consequences for the field of aneuploidy and cancer biology is based on comparison of two different trisomies. By the way, the authors have AF cells with trisomy of chromosome 18 and 21, but do not provide any data on the missegregation there. They only use them to show the levels of spartin and other proteins (Figure 4).

2) By FISH the authors observe missegregation basically only of chromosome 7 and/or 13, however, the chromosome count clearly show that there must be more chromosome missegregated. The authors just hand wave this. Moreover, chromosome 7 is not missegregated in AF+13, although it is frequently affect in DLD1+13. Why is that? Could the cell type of DLD1, colorectal cancer, affect the results, since the copy numbers of chromosome 7 and 13 are frequently altered in colorectal cancers?

---

## [Author Response]

*While both referees found your results interesting, they raised several concerns that are set out below. One referee commented that your mechanistic analysis of Spartin is interesting but based on two evidentiary leaps. First, that transient transfection of Spartin mimics the trisomic state, and second, that knockdown of Spartin mimics the disomic state. They suggested that you could strengthen this evidence using more specific techniques (lentiviral infection to introduce one copy of Spartin into the genome; CRISPR to remove one allele). These would undoubtedly strengthen your case but are substantial undertakings; therefore, in revising your study you should characterise your Spartin over expression and knock-down cell lines more carefully and discuss the potential caveats of your approaches*.

Although we realize that different experimental approaches have different strengths and weaknesses, we would like to point out here that we have now performed experiments to better characterize the role of Spartin in cytokinesis failure in trisomy 13 amniocytes (AF+13). By WB quantification, we find Spartin expression in AF+13 cells to be ∼1.5-fold that observed in AF cells, indicating that this system is much more comparable to the addition of a single copy of the gene. Our experiments in AF+13 cells fully support our previous findings in DLD1+13 cells.

*Major points*:

*1) The findings described here are at odds with the findings by*
[72]*. This study reported minimal chromosomal instability among aneuploid cells with eight different karyotypes. Only two different karyotypes are examined in this paper. This raises the question of the extent to which these results can be generalised*.

We have now included a more exhaustive discussion (in the subsection headed “Chromosomal instability in trisomic cells”) of our Results in light of the study by [72]. Briefly, we would like to summarize here a few points: (i) Valind et al. found an increase of (CIN)/somatic mosaicism for certain chromosomes in cells with certain karyotypes, and this information can be found in a supplemental table of the paper. Yet, the chromosomal instability they observed does not match that observed in CIN cancer cells (which is also the case in our study). However, this should not be surprising as the degree of CIN may scale up with the degree of aneuploidy. (ii) The cell type used in the Valind et al. study may also partly explain the difference between their data and other studies, such as studies using peripheral blood lymphocytes of Turner and Down syndrome patients (49; 50). (iii) Although Valind et al. analyzed more karyotypes, we examined more chromosomes and used several different approaches, as opposed to the single experimental approach used by Valind et al. We believe that a major reason for the discrepant conclusions reached by studies published over the years may be the difference in approaches used to assess CIN. When looking at cells undergoing mitosis (as we did in this study), one can directly assess the mis-segregation rates. However, mis-segregating cells may or may not divide further. Cell types in which mis-segregation does not cause a cell cycle arrest will “amplify” the aneuploidy by producing aneuploid daughter cells that will, in turn, keep dividing. In this case, using interphase FISH would be good enough to determine that aneuploidy promotes chromosome mis-segregation. However, if the cells produced after a mis-segregation event stop dividing, interphase FISH will largely underestimate the ability of aneuploidy to promote chromosome mis-segregation. In more extreme cases, mis-segregation could even result in cell death, and this again would result in very low levels of CIN as measured by interphase FISH even in the presence of high rates of chromosome mis- segregation. Thus, cell type-specific differences in aneuploidy-induced cell cycle arrest or cell death may explain the discrepancies between different studies. We believe that one strength of our study is the use of several different approaches to investigate the role of aneuploidy in causing chromosome mis-segregation and karyotipic heterogeneity/CIN. The points summarized here are now extensively discussed in the manuscript (please see the Discussion section).

*2) Two phenotypes of aneuploid cells are described in this manuscript: cytokinesis failure and the generation of lagging chromosomes. Are these two phenotypes related or distinct? Tetraploid cells are known to be chromosomally-unstable (e.g., Fujiwara, 2005). Are the lagging chromosomes in*
Figure 2
*occurring specifically in tetraploid cells? This could be tested by sorting a 2N population from each cell line and looking for laggards. FiFISH staining of chr. 7, 12 and 18 on interphase nuclei. How often are multipolar mitoses (which would be expected with the high frequency of tetraploidy) observed*?

Whereas it has been previously shown that tetraploid cells display high degrees of CIN (14; 21), and that multipolarity promotes formation of merotelic kinetochore attachments and hence anaphase lagging chromosomes (22; 61), we think that the two phenotypes we observe in our study are independent of each other. We believe this to be the case for two reasons: first, we do not observe multipolar mitoses (data now added in Figure 2—figure supplement 1); second, we find increased rates of anaphase lagging chromosomes also in DLD1+7, where tetraploidy is rare. We realize that the lack of multipolar mitoses in DLD1+13, where a significant fraction of the population is tetraploid may be surprising, as it was to us. However, this is in agreement with unpublished data from our lab in which we found normal centrosome numbers (1 centrosome/2 centrioles in G1, 2 centrosomes/2 centriole pairs in mitosis) in tetraploid cell clones generated by experimentally inhibiting cytokinesis. We now present the data on multipolar mitoses in the Results (in the subsection headed “Increased chromosome mis-segregation in cells with trisomy 7 or 13”; Figure 2—figure supplement 1) and discuss the points summarized here in the Discussion section.

*3) How was the tetraploidy/near tetraploidy determined? In*
Figure 3*, there appear to be more tetraploid wild-type AF cells than tetraploid AF +13 cells. However, in*
Figure 3*, the opposite is shown.*
Figure 3
*shows that at least 5 % of AF+13 cells are tetraploid/near tetraploid without explaining how were these number obtained. In the text it appears that this is extrapolation from the FISH staining of chr. 7, 12 and 18 on interphase nuclei. Identification of four signals per chromosome does not provide sufficient evidence of tetraploidy*.

We realize that this point was not well explained in the previous version of the manuscript. We have now clarified (mainly in the figure legend) that, as the reviewers pointed out, the data in Figure 3 are a quantification of interphase FISH with probes specific for three different chromosomes. The reviewers are also correct in that chromosome counts did not reveal tetraploidy in AF+13. However, as we now explain in the text (in the subsections “Increased chromosome mis-segregation in cells with trisomy 7 or 13” and “Aneuploidy confers karyotype-dependent phenotypes”), this could easily be explained if tetraploid AF+13 cells became arrested in the cell cycle. If this were the case, then one would expect to see tetraploid chromosome numbers in interphase nuclei, but not in metaphase spreads. We believe that simultaneous detection of four signals per chromosome for three different chromosomes constitutes strong evidence of tetraploidy in AF+13 nuclei.

*4) How many cells from the population are really trisomic? In the original publication (*[69]*, Cancer Research), it was shown that only approximately 60 % of the cells DLD1 + 13 contain three copies of chromosome 13 (20% disomic, rest tetrasomic). In particular in the light of the main finding of the manuscript that aneuploidy facilitates further chromosomal instability it would be very useful to know the true karyotypic variability of the analyzed cell lines (in particular when specifically the trisomic chromosome should be affected). The authors should at least provide percentage of trisomic cells within the population and show for example by multicolor FISH or by SNP arrays that no other recurrent aneuploidy occurred during the passaging. Also, providing the passage number would be useful. In the Materials and methods it is described that three cases of constitutional trisomy 13 and three diploid controls were used in the study. Is the result presented in figures as AF and AF+13 an average of these three different cell populations, or is it one example*?

The reviewers are correct. But because of the tendency of DLD1+13 cells to lose one copy of chromosome 13, the cells were subcloned at the beginning of this study. The cell population used for this study was assessed by chromosome painting with a FISH probe specific for chromosome 13 and we found that 83.5% of the cells were trisomic, 2.5% were tetraploid, and 4% were diploid. These data are now included in Figure 1. Moreover, aCGH data collected at later passages indicated no recurrent aneuploidy, although at that point the trisomic cell population had already diminished, as expected also based on our chromosome count data (Figure 3). A thorough description of this characterization of DLD1+13 cells, as well as characterization of AF+13 cells and specific information of the passage number at which various experiments were performed are now provided in the Results section, Figure 1, and Figure 1—figure supplement 1).

Moreover, we have now included information on AF cells in Table 1. The results presented for AF and AF+13 are in all cases averages of the three different cell populations listed in Table 1.

*5) The overexpression levels of protein Spartin should be quantified from three independent experiments. Further investigation into the importance of Spartin overexpression on cytokinesis failure is important as*
Figure 4
*demonstrates that Spartin is substantially over-expressed in DLD+13. Thus, Spartin may be upregulated in part as a general consequence of aneuploidy, and not simply because of its increased copy number. Alternatively, is the region 13q.13.3 amplified in DLD1+13*?

We have now included quantification of western blot data from three independent experiments (DLDs) or samples (AFs) (Figure 4). The amount of Spartin in DLD1+7 argues against a possible overexpression of Spartin as a general consequence of aneuploidy. Moreover, we also found that Spartin was not overexpressed in amniocytes carrying other trisomies (data now shown in Figure 4–figure supplement 5). Finally, the aCGH data (Figure 1—figure supplement 1) shows no evidence of amplification of region 13q13.3.

*6) The link to spastin, although intriguing, is not sufficiently substantiated to explain the observed phenotypes. How does the spartin/spastin pathway explain the increased rates of chromosome missegregation? An experiment showing that the overexpression of SPG20 in DLD1 leads to loss of spastin in midbody would be useful*.

It is possible that this was not clearly explained in the previous version of the manuscript, but we are not suggesting that the two different phenotypes described here (anaphase lagging chromosomes and cytokinesis failure) are both induced by mis- regulation of the spartin/spastin pathway. In fact, high rates of anaphase lagging chromosomes were also observed in DLD1+7 cells, which did not display spartin overexpression. We have revised the text throughout and we think it is now clear that, based on our data, we conclude that aneuploidy, regardless of the identity of the aneuploid chromosome, can promote chromosome mis-segregation. Our second conclusion is that different aneuploidies confer different phenotypes. In our study, we identified one such karyotype-phenotype connection by identifying a link between trisomy 13 and cytokinesis failure. To better get this point across, we have also slightly changed the title of the manuscript from “Aneuploidy causes karyotype-dependent phenotypes in human cells” to “Aneuploidy causes chromosome mis-segregation and karyotype-dependent phenotypes in human cells.” Finally, we think that the experiment suggested by the reviewers would have been a nice addition to our study, and therefore we attempted it. However, the low efficiency of transfection combined with the low number of cells in cytokinesis undermined the success of this experiment.

*In addition to these points the referees suggested that some clarification of the text will be required to address some issues as follows*:

*The authors' conclude that aneuploidy induces CIN. However, the cytokinesis block experiments in*
Figure 3
*actually argue against this statement. The authors report that only 1 out of 11 euploid chromosomes display frequent missegregation, while 3 out of 3 aneuploid chromosomes display frequent missegregation. This suggests that aneuploidy does not induce a general state of CIN, but instead aneuploid cells have particular trouble segregating aneuploid chromosomes. This discovery is certainly very interesting and worthy of further exploration. However, it undercuts the conclusion that aneuploidy causes CIN*.

We now discuss our data more extensively (in the subsection entitled “Increased chromosome mis-segregation in cells with trisomy 7 or 13”) and point out that although our cytokinesis-block assay (Figure 3) showed chromosome mis- segregation to be limited to certain chromosomes, the chromosome count data (Figure 3) suggest that chromosome mis-segregation must be more widespread than the cytokinesis-block assay reveals. For instance, the karyotypic heterogeneity observed in DLD1+7 cells, with chromosome numbers ranging between 42 and 53 (Figure 3), cannot be explained by exclusive mis-segregation of chromosome 7 (Figure 3), because this would produce a narrower chromosome number distribution around the modal number of 47.

*The authors claim that:* “*…addition to previously described mechanisms, [these findings] might explain the significantly higher rates of anaphase…*” *(in the Discussion). This implies that they found a new mechanism. However, no new mechanism of chromosome segregation errors is proposed, in particular if they state that there are no multipolar mitoses*.

The reviewers are correct that the wording used in our Discussion was inaccurate. The Discussion has been extensively revised and this statement no longer appears in the text.

[Editors' note: further revisions were requested prior to acceptance, as described below.]

*1) Somehow it is difficult to accept the general conclusion—*“*Aneuploidy causes chromosome mis-segregation*”*—when it is made on the basis of analysis of two different trisomies in only one cancerous and p53 deficient cell line (AF cells only with trisomy 13 were analyzed). Thus, this sweeping conclusion with potentially large consequences for the field of aneuploidy and cancer biology is based on comparison of two different trisomies. By the way, the authors have AF cells with trisomy of chromosome 18 and 21, but do not provide any data on the missegregation there. They only use them to show the levels of spartin and other proteins (*Figure 4*)*.

We have addressed this concern by making a number of changes, starting with the title, which now reads: “Chromosome mis-segregation and cytokinesis failure in trisomic human cells.” We also toned down our Discussion by specifying that the conclusion on chromosome mis-segregation is limited to the trisomies studied here (see first line of the Discussion). We also acknowledged the caveats related to the limited number of trisomies and limited number of chromosomes analyzed in our FISH experiments (see end of the subsection headed “Increased chromosome mis-segregation in cells with trisomy 7 or 13”). We finally acknowledged that “certain aneuploidies may not be sufficient to induce chromosome mis-segregation” (please see “Chromosomal instability in trisomic cells”).

*By the way, the authors have AF cells with trisomy of chromosome 18 and 21, but do not provide any data on the missegregation there. They only use them to show the levels of spartin and other proteins (*Figure 4*)*.

Although we have access to AF cells with other trisomies, we would like to point out that the experiments we performed (including several based on live-cell imaging) are technically very challenging in amniocytes. This may explain why other studies investigating the link between aneuploidy and CIN in primary cells employ interphase FISH as the only method of analysis. We feel that by using multiple approaches we can gain a deeper knowledge. However, analyzing other trisomies at this stage may delay the publication of this work by one or two years. Although we agree that it will be interesting to study these other trisomies and plan to do so, we feel the work included in this manuscript constitutes a complete story and its publication should not be delayed further.

*2) By FISH the authors observe missegregation basically only of chromosome 7 and/or 13, however, the chromosome count clearly show that there must be more chromosome missegregated. The authors just hand wave this*.

We feel that we did not hand wave this point, as we had discussed this both in the Results section and in the Discussion. However, we realize that discussing this point in two different places may have caused it to blend in the text and not stand out. So, we removed the discussion from the Results section and expanded our discussion of this issue in the Discussion section.

*Moreover, chromosome 7 is not missegregated in AF+13, although it is frequently affect in DLD1+13. Why is that? Could the cell type of DLD1, colorectal cancer, affect the results, since the copy numbers of chromosome 7 and 13 are frequently altered in colorectal cancers*?

We discuss now the distinct chromosome 7 mis-segregation rates found in DLD1+13 and AF+13 as suggested by the reviewer (please see the Discussion: “On the other hand, the finding that chromosome 7 mis-segregated in DLD1+13 […]a cell type-specific effect is plausible”).